# FedScar: Correcting Geometric Bias for Flatness-Consistent Federated Learning

**Jianfeng Lu** [1]   **Yuzhao Xiang** [1]   **Yue Chen** [2] *   **Gang Li** [3]   **Shuqin Cao** [4]   **Guanghui Wen** [5] *

## Abstract

Federated Learning (FL) often suffers from degraded generalization under statistical heterogeneity, where client updates systematically deviate from the global objective. While recent Sharpness-Aware Minimization (SAM) methods promote locally flat solutions, they implicitly assume that local flatness transfers to the global model, which generally does not hold under heterogeneous data distributions. This mismatch gives rise to a flatness discrepancy induced by misaligned loss landscapes. To address this issue, we propose FedScar, a federated optimization framework that explicitly corrects heterogeneity-induced geometric inconsistency. FedScar maintains a history-accumulated geometric bias to capture persistent curvature skew across clients, and employs a variance-aware injection mechanism to steer local updates toward regions that are flat with respect to the global objective. We provide a theoretical interpretation of FedScar as a Split-Dual ADMM formulation, which jointly enforces parameter consensus and geometric alignment. Extensive experiments under severe heterogeneity demonstrate that FedScar consistently reduces flatness discrepancy and improves generalization over state-of-the-art methods, without incurring additional communication overhead. The source code is available at https://github.com/Ultraman6/FedScar.

[1]School of Computer Science and Technology, Wuhan University of Science and Technology, China [2]Hubei Province Key Laboratory of Intelligent Information Processing and Real-time Industrial System, Wuhan University of Science and Technology, China [3]College of Computer Science, Inner Mongolia University, China [4]Key Laboratory of Social Computing and Cognitive Intelligence (Dalian University of Technology), Ministry of Education, China [5]School of Automation, Southeast University, China. Correspondence to: Yue Chen <chenyue@wust.edu.cn>, Guanghui Wen <ghwen@seu.edu.cn>.

*Proceedings of the $43^{rd}$ International Conference on Machine Learning*, Seoul, South Korea. PMLR 306, 2026. Copyright 2026 by the author(s).

## 1. Introduction

Federated Learning (FL) has established itself as a cornerstone paradigm for privacy-preserving machine learning, enabling distributed edge devices to collaboratively train a global model without exposing raw data (McMahan et al., 2017; Kairouz et al., 2021). Despite its promise, the practical deployment of FL is fundamentally challenged by *statistical heterogeneity* (Li et al., 2020), where the divergence between local and global data distributions causes client optimization trajectories to drift significantly from the global objective. While classic variance-reduction methods like SCAFFOLD (Karimireddy et al., 2020) effectively mitigate first-order parameter drift to accelerate convergence, they largely overlook the critical second-order landscape properties that govern generalization. Recent studies (Solans et al., 2024; Sun et al., 2023b) reveal that under severe heterogeneity, standard federated averaging tends to converge to sharp minima characterized by with high curvature and is extremely sensitivity to weight perturbations. This geometric fragility leads to catastrophic performance degradation when the global model is deployed to unseen clients or novel distributions, marking a critical bottleneck for robust federated optimization.

To address this generalization collapse, Sharpness-Aware Minimization (SAM) (Foret et al., 2021a) and its federated variants (e.g., FedSAM (Qu et al., 2022)) have been introduced to simultaneously minimize loss value and loss sharpness. However, these methods rely on a perilous implicit assumption: that *local flatness* achieved on private data automatically transfers to *global flatness*. We argue that under non-IID distributions, this assumption fundamentally breaks down. Since each client optimizes for its own local basin, the aggregated global model often lands on a sharp ridge or saddle point sandwiched between disjoint local optima, rather than in a truly flat global basin. We term this structural misalignment the **Flatness Discrepancy**. Existing solutions either ignore this discrepancy, resulting in suboptimal generalization, or attempt to fix it by transmitting heavy auxiliary variables or Hessian approximations (e.g., FedLESAM (Fan et al., 2024a)), incurring prohibitive communication and computation overheads that negate the efficiency benefits of FL.

How do we resolve this discrepancy without imposing extra

system costs? Our central insight is that the divergence between local and global optimal perturbation directions is not merely ephemeral stochastic noise, but reflects a **structural, accumulable geometric bias** induced by persistent data heterogeneity. Unlike random variance which can be averaged out over rounds, this geometric bias follows a traceable trajectory embedded in the optimization history, systematically steering the model towards sharp regions. Consequently, we posit that Flatness Discrepancy should be modeled as a *learnable latent variable*. This perspective transitions the optimization paradigm from **stochastic variance mitigation** to **structural bias rectification**, allowing us to correct local optimization directions using historical information to precisely align with the global geometry.

Guided by this perspective, we propose **FedScar** (**Fed**erated **S**harpness and **C**ontrol v**ar**iate **R**egularization), a novel framework that transcends standard local flatness seeking. Formulated via a Split-Dual ADMM approach, FedScar explicitly maintains a history-accumulated dual variable to capture and rectify the persistent geometric skew. By injecting this estimated bias into the local objective, FedScar dynamically steers client updates toward regions that satisfy both local flatness and global alignment, effectively **modulating** the local landscape topology to match the global objective. Furthermore, a variance-aware update rule adaptively regulates the correction intensity, ensuring stable convergence even under severe heterogeneity without incurring any additional communication overhead.

Our main contributions are summarized as follows:

- We propose FedScar, a novel framework that characterizes Flatness Discrepancy as a learnable geometric bias. By employing a history-accumulated dual-variable mechanism, FedScar simultaneously corrects first-order parameter drift and second-order geometric misalignment without extra communication costs, offering a scalable solution for heterogeneous FL.
- We provide a theoretical interpretation of FedScar under a Split-Dual ADMM (Alternating Direction Method of Multipliers) formulation, proving that it jointly optimizes global parameter consensus and geometric smoothness with a convergence rate of $\mathcal{O}(1/\sqrt{T})$, matching state-of-the-art non-convex rates.
- We conduct extensive experiments on CIFAR-10 and CIFAR-100 under diverse heterogeneity settings. Results demonstrate that FedScar consistently reduces flatness discrepancy and outperforms state-of-the-art methods, including FedSAM, FedGF, and FedLESAM, particularly in extreme non-IID scenarios.

**Conflict of Interest Disclosure.** The authors declare that they have no associated financial conflicts of interest or competing interests regarding the publication of this paper.

## 2. Preliminaries and Motivation

### 2.1. Federated Optimization and Heterogeneity

We consider a standard Federated Learning setting with $N$ clients. The goal is to minimize the global empirical risk function $f(w)$: $\min_{w \in \mathbb{R}^d} f(w) \triangleq \sum_{i=1}^N p_i f_i(w)$, where $p_i$ is the weight of client $i$ (usually $p_i = \frac{|D_i|}{|D|}$), and $f_i(w) = \mathbb{E}_{\xi \sim D_i}[\ell(w; \xi)]$ is the local expected loss on the private dataset $D_i$. In standard FedAvg (McMahan et al., 2017), clients perform multiple steps of Stochastic Gradient Descent (SGD) locally to minimize $f_i(w)$. However, due to the statistical heterogeneity (Non-IID data), the local data distribution $D_i$ differs from the global distribution $D$. Consequently, the local optimum $w_i^* = \arg\min f_i(w)$ diverges from the global optimum $w^* = \arg\min f(w)$. This discrepancy leads to the well-known *client drift* phenomenon (Karimireddy et al., 2020), where the aggregation of local updates $\sum p_i \nabla f_i(w)$ accurately estimates the global gradient $\nabla f(w)$ only when the model $w$ is synchronized. Once clients perform multiple local updates, the accumulated drift causes the global model to suffer from slow convergence and unstable optimization trajectories.

### 2.2. Sharpness-Aware Minimization in FL

Beyond convergence speed, the generalization capability of the trained global model is of paramount importance. Classical Empirical Risk Minimization (ERM) in FL tends to converge to sharp minima, which are sensitive to data perturbations and generalize poorly to unseen clients (Caldarola et al., 2022). To address this, Sharpness-Aware Minimization (SAM) (Foret et al., 2021a) aims to minimize both the loss value and the loss sharpness. The global SAM objective is defined as: $\min_w \mathcal{L}^{SAM}(w) \triangleq \max_{\|\epsilon\|_2 \leq \rho} f(w + \epsilon) \approx f(w) + \max_{\|\epsilon\|_2 \leq \rho} \epsilon^\top \nabla f(w)$, where $\rho$ is the perturbation radius. Existing works like FedSAM (Qu et al., 2022) directly apply SAM to local objectives, solving the following surrogate problem: $\min_w \sum_{i=1}^N p_i \max_{\|\epsilon_i\|_2 \leq \rho} f_i(w + \epsilon_i)$. In this formulation, each client $i$ computes a local optimal perturbation $\epsilon_i^*(w) \approx \rho \frac{\nabla f_i(w)}{\|\nabla f_i(w)\|}$ based on its local geometry to perform updates.

### 2.3. From Flatness Discrepancy to Geometric Bias

While FedSAM (Qu et al., 2022) introduces seeking flat minima into FL, it overlooks the geometric misalignment arising from heterogeneity. Ideally, perturbations should align with the *global* gradient ($\epsilon^* \propto \nabla f(w)$). However, clients compute perturbations using *local* gradients ($\epsilon_i^* \propto \nabla f_i(w)$). Due to non-IID data, $\epsilon_i^*$ diverges from $\epsilon^*$, creating a *gradient approximation bias* where aggregated updates fail to target the global flat minimum. To quantify this, we analyze the *Flatness Discrepancy* $\Delta_{\mathcal{F}}$ (Lee & Yoon, 2024)

Table 1. Comparison of different algorithms.

| Research work | Flatness | Flatness Discrepancy | Bytes | FLOPS |
|---|---|---|---|---|
| FedSAM | ✓ | ✗ | 1× | 2× |
| FedSpeed | ✓ | ✗ | 1× | 2× |
| FedSMOO | ✓ | ✗ | 2× | 2× |
| FedLESAM | ✓ | ✗ | 1× | 1× |
| FedGMT | ✓ | ✗ | 1.5× | 1.33× |
| FedScar(Ours) | ✓ | ✓ | 1.5× | 1× |

and decompose it into two consistency components:

$$\Delta_{\mathcal{F}} = \left| \underbrace{\left( \max_{\|\epsilon\|} f(w+\epsilon) - \sum_{i=1}^{N} p_i \max_{\|\epsilon_i\|} f_i(w+\epsilon_i) \right)}_{\textbf{Perturbed Loss Discrepancy}} \right.$$

$$\left. - \underbrace{\left( f(w) - \sum_{i=1}^{N} p_i f_i(w) \right)}_{\textbf{Original Loss Discrepancy}} \right|. \quad (1)$$

Since the FL objective implies $f(w) \approx \sum p_i f_i(w)$, the **Original Loss Discrepancy** vanishes. The discrepancy is thus dominated by the **Perturbed Loss Discrepancy**, proving that consistency in the original loss does not guarantee consistency in the perturbed landscape.

We attribute this perturbed discrepancy to the **Geometric Bias** $\mathcal{B}_i^{geo}(w) \triangleq \epsilon_i^* - \epsilon^*$. By Taylor expansion, the divergence in the update direction is approximated as:

$$\nabla f_i(w + \epsilon_i^*) \approx \nabla f_i(w + \epsilon^*) + \underbrace{\nabla^2 f_i(w)\mathcal{B}_i^{geo}(w)}_{\text{Geometric Drift}}. \quad (2)$$

This derivation reveals that standard updates are corrupted by a *Geometric Drift* term dependent on local Hessians and heterogeneity. Crucially, this bias is structural rather than random, exhibiting temporal correlations that necessitate a history-accumulated correction mechanism.

**Empirical Flatness Discrepancy.** As computing optimal perturbations is intractable, we propose an efficient empirical metric to monitor discrepancy. Inspired by (Li et al., 2025a), we quantify the *Empirical Flatness Discrepancy* $\tilde{\Delta}_{\mathcal{F}}^{(t)}$ by comparing the loss of the aggregated model against the local models:

$$\tilde{\Delta}_{\mathcal{F}}^{(t)} \triangleq \left| \sum_{i=1}^{N} p_i f_i(w^t) - \sum_{i=1}^{N} p_i f_i(w_i^{(t)}) \right|. \quad (3)$$

Intuitively, a large $\tilde{\Delta}_{\mathcal{F}}^{(t)}$ indicates that the aggregated model $w^t$ sits on a sharp peak, while a small value implies convergence to a consistent, flat basin. We rigorously link this metric to Hessian-based sharpness in Appendix B. FedScar does not attempt to recover the exact global perturbation direction. Instead, it estimates a persistent bias subspace that correlates with heterogeneity-induced curvature skew.

To validate our motivation, Figure 1 analyzes the correlation between landscape metrics and test accuracy. While absolute Flatness (Fig. 1b) shows negligible correlation (Pearson r=0.009) with performance, Flatness Discrepancy (Fig. 1a) exhibits a strong negative correlation ($r = -0.716$). This stark contrast serves as empirical evidence that optimizing

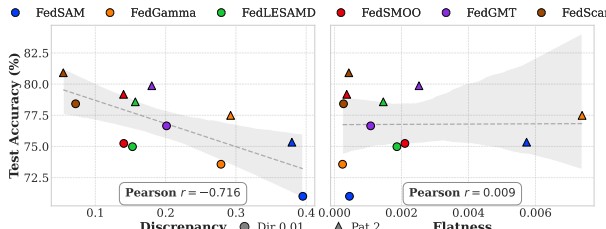

Figure 1. **Correlation between generalization performance and Flatness Discrepancy.** The experiment is conducted on CIFAR-10 under a highly heterogeneous setting (Dirichlet distribution with $\alpha = 0.01$ and label imbalance ratio 0.5).

global sharpness is not equivalent to minimizing discrepancy: as shown in Fig. 1b, models can achieve desirable low sharpness yet yield poor generalization, confirming that Flatness Discrepancy captures a distinct geometric misalignment that standard sharpness metrics fail to detect. Unlike baselines (e.g., FedSAM, FedGamma) that suffer from severe misalignment, **FedScar** consistently achieves the lowest discrepancy and highest accuracy. Finally, Table 1 summarizes that while most methods ignore flatness discrepancy or incur high overheads (e.g., 2× FLOPS in FedSMOO), FedScar uniquely optimizes geometric alignment with minimal communication (1.5× Bytes) and no computational overhead (1× FLOPS).

## 3. Related Work

**Statistical Heterogeneity and Client Drift.** Statistical heterogeneity constitutes a fundamental hurdle in FL, inducing client drift where local updates diverge from the global objective (Karimireddy et al., 2020). Classic solutions like FedProx (Li et al., 2020) mitigate this via proximal constraints, while advanced optimizers such as SCAFFOLD (Karimireddy et al., 2020) and FedDyn (Acar et al., 2021) explicitly correct drift using control variates and dynamic regularization. Recently, FedInit (Sun et al., 2023b) proposed a stage-wise relaxed initialization strategy to revise local divergence without extra communication costs. However, these approaches primarily focus on first-order parameter drift, overlooking the critical second-order geometric inconsistency. FedScar distinguishes itself by employing a unified dual-variable framework to simultaneously rectify both parameter drift and geometric bias.

**Flat Minima Seeking in FL.** Seeking flat minima has emerged as a promising direction to enhance generalization in FL. Adapting Sharpness-Aware Minimization (SAM) (Foret et al., 2021b), algorithms like FedSAM (Qu et al., 2022) and FedASAM (Caldarola et al., 2022) encourage clients to find flat minima within their local neighborhoods. Despite improving local flatness, recent studies identify a critical flatness discrepancy (Lee & Yoon, 2024; Fan et al., 2024b), where locally flat landscapes fail to translate to global flatness due to significant distribution shifts. Unlike

these methods that optimize local geometry in isolation, FedScar acknowledges this disconnect and introduces a mechanism to explicitly rectify local perturbation directions, ensuring alignment with the global loss landscape.

**Flatness Consistency in FL.** To ensure flatness consistency across clients, recent efforts attempt to incorporate global sharpness information. Methods like FedGamma (Dai et al., 2024) and FedSMOO (Sun et al., 2023a) synchronize curvature via global momentum or ADMM, while FedLESAM (Fan et al., 2024b) and FedGMT (Li et al., 2025b) utilize historical trajectories to approximate global perturbations. However, these strategies often incur high communication overheads or rely on implicit regularization that falters under extreme heterogeneity. In contrast, FedScar models geometric mismatch as a learnable, history-accumulated bias. By correcting curvature misalignment via a cost-effective, variance-aware mechanism, FedScar achieves superior global consistency without imposing additional uplink communication burdens.

# 4. Methodology

We propose FedScar, a Split-Dual ADMM framework designed to explicitly minimize Flatness Discrepancy by decoupling parameter drift from geometric drift. Unlike previous approaches that treat these errors indistinguishably, FedScar enforces both first-order parameter consistency and second-order geometric consistency through a history-accumulated bias correction mechanism.

## 4.1. Split-Dual ADMM Formulation

To address the misalignment between local and global loss landscapes, we reformulate the federated optimization problem as a bi-level constrained optimization. We impose two distinct constraints: one for parameter alignment ($w_i = z$) and another for geometric alignment ($\mathcal{G}(w_i) \approx \mathcal{G}(z)$), where $\mathcal{G}(\cdot)$ denotes the geometric property such as the local curvature direction. To solve this, we construct a generalized Augmented Lagrangian by splitting the dual variables into two vectors: $h_i$ (Drift Dual) to manage the parameter constraint, and $b_i$ (Bias Dual) to manage the geometric constraint. The formulation is defined as:

$$\mathcal{L}_{\text{FedScar}} = \sum_{i=1}^{N} p_i \bigg( f_i(w_i) + \frac{\beta}{2} \|w_i - z\|^2$$
$$+ \beta \langle w_i - z, h_i \rangle + \langle w_i - z, \phi(b_i) \rangle \bigg), \quad (4)$$

where $\beta$ is a penalty parameter and $\phi(b_i)$ maps the geometric bias. Here, $h_i$ acts as the standard control variate to correct mean drift, while $b_i$ serves as a dedicated corrector for the geometric bias $\mathcal{B}_i^{geo}$. Explicitly separating these dual roles decouples the minimization of the loss function from the regulation of landscape curvature, ensuring that parameter consensus and geometric alignment are satisfied simultaneously without optimization conflicts. FedScar solves this min-max saddle point problem via alternating Primal Minimization during local training and Dual Ascent during server aggregation.

## 4.2. Heterogeneity-Aware Geometric Bias Estimation

A core innovation of FedScar is the adaptive estimation of the geometric bias $b_i$. Since the intensity of geometric misalignment fluctuates with data heterogeneity, we introduce a Heterogeneity-Sensitive Perturbation Scale $\Sigma^{(t)}$ to dynamically control the correction strength. At the end of round $t$, the server quantifies system-wide heterogeneity by calculating the weighted dispersion of local updates $\Delta w_i^{(t)}$ relative to the global update $\Delta_{global}^{(t)} = w^{t+1} - w^t$:

$$\Sigma^{(t)} = \frac{\alpha}{N} \sum_{i \in \mathcal{A}^{(t)}} \|\Delta w_i^{(t)} - \Delta_{global}^{(t)}\|_2, \quad (5)$$

where $\mathcal{A}^{(t)}$ is the set of active clients and $\alpha$ is a scaling hyperparameter. This metric serves as an indicator of the global landscape properties; a larger value implies significant trajectory divergence among clients, suggesting a sharp global landscape that necessitates increased geometric correction intensity.

Leveraging this scale, we employ a Variance-Weighted Update Mechanism to estimate $b_i$. The server utilizes the drift direction vector $v_i^{(t)} = \Delta_{global}^{(t)} - \Delta w_i^{(t)}$ scaled by $\Sigma^{(t)}$:

$$b_i^{(t+1)} \leftarrow \gamma b_i^{(t)} + \Sigma^{(t)} \cdot \frac{v_i^{(t)}}{\|v_i^{(t)}\|_2 + \epsilon}, \quad (6)$$

where $\gamma \in [0, 1)$ is a forgetting factor. This update rule ensures that $b_i$ incorporates both the precise orientation ($v_i/\|v_i\|$) and the severity ($\Sigma^{(t)}$) of historical drifts. Unlike instantaneous gradient estimates susceptible to stochastic noise, this accumulation mechanism acts as a low-pass filter, isolating the structural geometric bias inherent to the local data distribution.

Functionally, $b_i$ serves as a geometric memory that encodes subspaces exhibiting curvature inconsistency between local and global objectives. By accumulating the divergence vector $v_i$, $b_i$ identifies trajectories where local minima deviate from the global consensus. Consequently, the term $-\langle w, b_i \rangle$ imposes a directional penalty, elevating the optimization landscape along these conflicting paths to constrain the solution space toward basins that simultaneously satisfy local flatness and global geometric alignment.

## 4.3. Consistency-Regularized Optimization Loop

With the estimated bias, FedScar executes a consistency-regularized optimization loop comprising local training and

global closure.

**Local Optimization.** Client $i$ initializes $w_{i,0} = w^t$ and minimizes the specific FedScar objective:

$$
\begin{aligned}
\min_w \tilde{f}_i(w) \triangleq &f_i(w) + \frac{\beta}{2}\|w - w^t\|^2 \\
&+ \beta\langle w, h_i\rangle + \langle w, \bar{b} - b_i\rangle.
\end{aligned}
\tag{7}
$$

The gradient used for SGD is $g_{i,k} = \nabla f_i(w_{i,k}) + \beta(w_{i,k} - w^t) + \beta h_i + (\bar{b} - b_i)$. Here, the term $(\bar{b} - b_i)$ acts as a geometric counter-force. Mathematically, this term alters the local optimization landscape, redirecting the trajectory from sharp local minima towards regions that are conducive to global aggregation, thereby reducing the flatness discrepancy.

**Global Aggregation and Closure.** After $K$ local steps, the server performs a two-stage aggregation to close the ADMM loop. First, the drift dual $h_i$ is updated via dual ascent to capture the accumulated parameter drift: $h_i^{(t+1)} \leftarrow h_i^{(t)} + \frac{1}{\eta_g K}(w^t - w_i^{(t+1)})$. Second, the server updates the global model $z$ (denoted as $w^{t+1}$) by solving $\nabla_z \mathcal{L}_{\text{FedScar}} = 0$:

$$
w^{t+1} \leftarrow w_{agg} + \eta_g \beta \frac{1}{N}\sum_{j=1}^{N}(h_j - b_j).
\tag{8}
$$

This step completes the optimization loop by integrating the collective geometric information encoded in the dual residuals, ensuring the global model evolves towards the intersection of local flat regions.

# 5. Theoretical Analysis

In this section, we analyze the convergence of global criterion and flatness discrepancy on FedScar.

## 5.1. Convergence Analysis

we analyze the convergence properties of FedScar by the theoretical frameworks of recent Federated SAM studies (Qu et al., 2022; Li et al., 2025a), we establish that FedScar achieves a convergence rate of $\mathcal{O}(1/\sqrt{T})$ under non-convex settings, while effectively mitigating the error terms induced by geometric heterogeneity. To facilitate the analysis, we adopt the following standard assumptions regarding the objective functions and data heterogeneity.

**Assumption 5.1 (L-Smoothness).** The local objective functions $f_i(w)$ and the global objective $f(w)$ are differentiable and $L$-smooth. Specifically, for all $x, y \in \mathbb{R}^d$ and $i \in \{1, \ldots, N\}$, the inequality $\|\nabla f_i(x) - \nabla f_i(y)\| \leq L\|x - y\|$ holds. This assumption implies that the Hessian matrices are bounded, i.e., $\|\nabla^2 f_i(w)\| \leq L$.

**Assumption 5.2 (Unbiased Gradient and Bounded Variance).** The stochastic gradient oracle $g_i(w; \xi)$ accessed by client $i$ serves as an unbiased estimator of local full gradient, satisfying $\mathbb{E}_\xi[g_i(w; \xi)] = \nabla f_i(w)$. Furthermore, the variance of the stochastic gradient is bounded by a non-negative constant $\sigma^2$, such that $\mathbb{E}_\xi[\|g_i(w; \xi) - \nabla f_i(w)\|^2] \leq \sigma^2$.

**Assumption 5.3 (Bounded Heterogeneity).** The statistical heterogeneity among clients is bounded. We assume there exist constants $G \geq 0$ and $H \geq 0$ such that for all $w$, the gradient dissimilarity satisfies $\frac{1}{N}\sum_{i=1}^{N}\|\nabla f_i(w) - \nabla f(w)\|^2 \leq G^2$, and the Hessian dissimilarity satisfies $\frac{1}{N}\sum_{i=1}^{N}\|\nabla^2 f_i(w) - \nabla^2 f(w)\|^2 \leq H^2$. The latter ensures that the geometric bias remains controllable.

**Assumption 5.4 (Bounded Perturbation Radius).** The perturbation radius $\rho$ used in the local SAM updates is chosen to be sufficiently small. Specifically, we require $\rho \leq c/L$ for some constant $c < 1$. This condition ensures that the perturbed loss landscape remains within the valid region of the first-order Taylor approximation, limiting the approximation error of the perturbation direction.

**Remark.** Bounding Hessian dissimilarity is indispensable for distributed curvature optimization (Safaryan et al., 2022; Crane & Roosta, 2022). By Taylor expansion, the divergence between local and global SAM updates decomposes into gradient drift and second-order perturbation drift, making cross-client Hessian bounds strictly necessary to control the latter. Furthermore, this assumption avoids unrealistic dimension-dependent penalties. Because SAM penalizes the spectral norm rather than the trace (Wen, 2023) and inherently induces low-rank curvature structures (Andriushchenko & Flammarion, 2023), the empirical Hessian variance remains exceptionally small and fundamentally dimension-free. These theoretical properties rigorously ground our subsequent convergence guarantees.

**Theorem 5.5 (Convergence of FedScar).** *Let Assumptions 5.1 through 5.4 hold. Suppose the learning rate satisfies $\eta \leq \frac{1}{8LK}$, where $K$ is the number of local steps. For a total of $T$ communication rounds, the sequence of global models $\{w^t\}_{t=0}^{T}$ generated by FedScar satisfies:*

$$
\min_{t\in\{0,\ldots,T-1\}}\mathbb{E}\|\nabla f(w^t)\|^2 \leq \frac{4\Delta_F}{\eta KT} + \mathcal{O}\left(\frac{\sigma^2}{K}\right) + \mathcal{O}\left(\eta^2 K^2 \mathcal{E}_{drift}\right),
\tag{9}
$$

*where $\Delta_F = f(w^0) - f^*$ is the initial function value gap, and $\mathcal{E}_{drift} \triangleq G^2 + \sigma^2$, which explicitly accounts for the impact of data heterogeneity ($G^2$) and stochastic gradient variance ($\sigma^2$) on the client drift.*

**Remark.** The presence of the $\mathcal{O}(\eta^2 K^2 \mathcal{E}_{drift})$ term in Theorem 5.5 captures the raw accumulation of client drift prior to step-size substitution. This formulation mathematically isolates the structural advantage of FedScar. Standard analyses of FedSAM often report post-substitution bounds yielding an apparent $\mathcal{O}(\sqrt{K})$ dependence. However, under strictly aligned step-size scaling $\eta = \mathcal{O}(1/(K\sqrt{T}))$, the

term $\eta^2 K^2$ reduces to $\mathcal{O}(1/T)$, ensuring that the asymptotic convergence rate does not degrade with larger local steps $K$. The fundamental superiority of FedScar lies in the constant coefficient preceding the heterogeneity term $G^2$. While standard FedSAM accumulates uncorrected geometric drift scaling with $G^2$, FedScar actively cancels this bias via the history-accumulated dual variable $b_i$, leading to a strictly tighter convergence bound and enabling greater convergence efficiency.

## 5.2. Convergence Analysis of Flatness Discrepancy

To theoretically elucidate why FedScar surpasses the limitations of standard SAM-based methods, we derive a convergence bound for the Flatness Discrepancy $\Delta_{\mathcal{F}}$. Unlike prior works (e.g., FedGF (Lee & Yoon, 2024)) that bound this discrepancy by a static heterogeneity constant $\sigma_g^2$, our analysis explicitly models the dynamics of the bias correction variable $b_i$. We define the *Residual Geometric Bias* at round $t$ as $\Psi_t \triangleq \frac{1}{N} \sum_{i=1}^{N} \|\mathcal{D}_i(w^t) - (b_i^t - \bar{b}^t)\|^2$, where $\mathcal{D}_i(w^t) = \nabla F_i(w^t) - \nabla F(w^t)$ represents the gradient drift.

**Theorem 5.6 (Bias-Corrected Discrepancy Convergence).** *Under Assumptions 5.1-5.3, and assuming the dual variable $b_i$ is updated with a forgetting factor $\gamma \in (0, 1]$, the Flatness Discrepancy of FedScar at round $T$ satisfies:*

$$\Delta_{\mathcal{F}}(w^T) \leq \underbrace{\mathcal{O}\left(\frac{\sigma_l \sqrt{L}}{\sqrt{T}}\right)}_{\text{Optimization Noise}} + \underbrace{\rho \cdot \sqrt{\left(1 - \frac{\gamma}{2}\right)^T \sigma_g^2 + \frac{4L^2\eta^2\sigma_l^2}{\gamma}}}_{\text{Decaying Geometric Misalignment}},$$
(10)

*where $\eta$ is the learning rate, $\sigma_g^2$ is the initial data heterogeneity, and $\sigma_l^2$ is the stochastic gradient variance.*

**Remark.** The analysis of the second term in Eq. (10) reveals a critical distinction in convergence behavior for $\gamma \in (0, 1]$. For standard FedSAM, the discrepancy is dominated by a static, irreducible heterogeneity term $\rho \cdot \sigma_g$ derived directly from the base recursion. In stark contrast, FedScar explicitly introduces a geometric contraction factor $(1 - \gamma/2)^T$ to the heterogeneity variance $\sigma_g^2$. This theoretically proves that FedScar actively dissolves the impact of statistical heterogeneity over time. Consequently, the discrepancy asymptotically converges to a minimal noise floor governed strictly by optimization variance, successfully eliminating the persistent geometric misalignment inherent to uncorrected SAM.

## 6. Experiments

### 6.1. Experimental Setup

**Datasets and Models.** To adapt to different complexity levels, we employ a customized **CNN** architecture for CIFAR-10 and **ResNet-8** for CIFAR-100. Both architectures replace Batch Normalization with Group Normalization (GN) to mitigate statistical instability in non-IID settings. Further details regarding model architectures and dataset statistics are provided in Appendix F.

**Heterogeneous Partition Strategy.** We simulate realistic federated scenarios with severe heterogeneity in both label distribution and data quantity. Specifically, we partition data among $N = 100$ clients using a Dirichlet distribution $Dir(\alpha)$ to simulate feature skew, and apply a log-normal distribution to sample sizes to simulate quantity imbalance.

**Baselines and Implementation.** We compare FedScar against a broad spectrum of SOTA algorithms, categorized into two groups: ① *Traditional FL* focusing on client drift or statistical diversity (FedAvg, SCAFFOLD, FedDyn, and FedInit); and ② *SAM-based FL* targeting geometric flatness (FedSAM, FedGamma, FedSmoo, FedLesamd, and FedGMT). We simulate a federated system with 100 clients over 500 communication rounds, utilizing an SGD optimizer with momentum. Details are available in Appendix F.

### 6.2. Performance Evaluation

**Analysis on Best Generalization.** As detailed in Table 2, **FedScar** consistently establishes the highest generalization upper bound across varying complexities. While performance gaps are moderate under mild heterogeneity, FedScar distinctly dominates as statistical divergence intensifies. Specifically, on CIFAR-10 with extreme heterogeneity (Dir 0.01, $p = 10\%$), FedScar achieves a peak accuracy of $78.42\%$, outperforming FedAvg and FedSAM by over $7\%$. This advantage scales to the complex CIFAR-100 benchmark; under the highly skewed Pathological setting, FedScar ($49.67\%$) surpasses the state-of-the-art FedGMT ($48.57\%$) and significantly leads FedAvg ($39.90\%$). These results confirm that by explicitly correcting geometric bias, FedScar effectively navigates saddle points to locate superior optima even under severe data non-IIDness.

**Analysis on Generalized Stability.** Beyond peak performance, the mean accuracy (over the final 50 rounds) highlights convergence stability. Traditional baselines suffer from severe oscillation in heterogeneous settings; for instance, on CIFAR-10 (Dir 0.01), FedSAM collapses from a peak of $71.18\%$ to a mean of $63.42\%$, exposing its inability to stabilize against gradient noise. In stark contrast, **FedScar** maintains a minimal peak-to-mean gap (e.g., $78.42\%$ vs. $77.24\%$), demonstrating robust resistance to client drift. This resilience persists in CIFAR-100 (Pat 20), where FedScar sustains a mean accuracy of $48.57\%$, outperforming FedGMT ($47.92\%$) and FedSAM ($37.20\%$). The data verifies that FedScar's history-accumulated correction mitigates variance from sparse sampling ($p = 10\%$), anchoring the global model in flat, stable basins.

**Analysis on Efficiency.** Figure 2 illustrates the efficiency

*Table 2.* Test accuracy on CIFAR-10 and CIFAR-100 benchmarks. We report **Max Acc. / Last-50 Mean Acc.** under various heterogeneity settings (Dirichlet $\alpha = \{1.0, 0.1, 0.01\}$ and Pathological $K = \{10, 20\}$ classes/client) and participation rates $p \in \{10\%, 20\%\}$.

| Method | Dir 1.0 | | Dir 0.1 | | Dir 0.01 | | Pat 10 | | Pat 20 | |
|---|---|---|---|---|---|---|---|---|---|---|
| | $p = 10\%$ | $p = 20\%$ | $p = 10\%$ | $p = 20\%$ | $p = 10\%$ | $p = 20\%$ | $p = 10\%$ | $p = 20\%$ | $p = 10\%$ | $p = 20\%$ |
| Dataset: CIFAR-10 | | | | | | | | | | |
| FedAvg | 77.17 / 75.79 | 77.48 / 76.48 | 76.33 / 71.69 | 76.42 / 73.86 | 71.00 / 63.20 | 73.71 / 68.72 | 80.51 / 79.18 | 80.54 / 79.85 | 75.34 / 66.44 | 77.32 / 71.27 |
| Scaffold | 78.04 / 77.41 | 78.95 / 78.33 | 77.75 / 75.94 | 78.03 / 77.04 | 73.54 / 69.69 | 75.51 / 73.78 | 81.67 / 81.22 | 82.12 / 81.85 | 76.99 / 73.13 | 77.62 / 74.83 |
| FedDyn | 78.29 / 77.96 | 79.34 / 78.98 | 77.79 / 76.88 | 78.39 / 77.69 | 74.98 / 72.55 | 76.87 / 74.97 | 81.81 / 81.33 | 82.27 / 82.06 | 78.58 / 74.68 | 79.81 / 77.41 |
| FedInit | 77.85 / 77.12 | 78.41 / 77.65 | 77.03 / 75.91 | 78.20 / 76.62 | 74.26 / 71.83 | 77.07 / 74.10 | 81.07 / 80.55 | 81.68 / 81.14 | 77.33 / 75.04 | 80.02 / 77.14 |
| FedSAM | 77.97 / 76.62 | 78.29 / 77.34 | 76.38 / 72.29 | 76.93 / 74.39 | 71.18 / 63.42 | 73.73 / 68.80 | 81.20 / 79.89 | 81.43 / 80.69 | 75.94 / 67.07 | 77.53 / 72.18 |
| FedGamma | 79.02 / 78.34 | 79.46 / 78.96 | 77.95 / 76.34 | 78.58 / 77.61 | 73.57 / 69.97 | 75.71 / 73.93 | 82.46 / 81.93 | 82.68 / 82.40 | 77.49 / 74.41 | 78.16 / 75.86 |
| FedLesamd | 79.00 / 78.42 | 79.38 / 79.04 | 78.30 / 77.40 | 78.96 / 78.38 | 75.24 / 72.67 | 77.23 / 75.64 | 82.28 / 81.70 | 82.56 / 82.16 | 78.77 / 75.86 | 79.84 / 78.75 |
| FedSMOO | 79.60 / 79.04 | 80.67 / 80.04 | 78.85 / 77.71 | 78.99 / 78.41 | 75.54 / 72.81 | 77.97 / 76.26 | 82.88 / 82.52 | 82.69 / 82.51 | 79.18 / 77.01 | 80.45 / 79.32 |
| FedGMT | 80.31 / 79.83 | 81.70 / 80.82 | 80.17 / 79.49 | 80.17 / 80.18 | 76.65 / 75.28 | 77.99 / 77.19 | 83.84 / 83.26 | 84.02 / 83.81 | 79.87 / 78.28 | 79.51 / 78.35 |
| **FedScar** | **81.36 / 80.82** | **81.41 / 80.96** | **80.51 / 79.88** | **80.96 / 80.35** | **78.42 / 77.24** | **78.75 / 78.05** | **84.58 / 84.13** | **84.22 / 84.01** | **80.91 / 79.85** | **81.88 / 81.25** |
| Dataset: CIFAR-100 | | | | | | | | | | |
| FedAvg | 44.41 / 43.41 | 45.83 / 45.01 | 40.99 / 39.01 | 43.31 / 42.32 | 37.87 / 35.19 | 40.31 / 39.03 | 44.08 / 41.74 | 45.62 / 44.63 | 39.90 / 37.01 | 42.25 / 41.11 |
| Scaffold | 47.81 / 46.91 | 49.81 / 49.20 | 44.48 / 43.57 | 46.20 / 45.61 | 42.04 / 40.82 | 44.04 / 42.97 | 47.72 / 46.86 | 44.57 / 43.60 | 46.65 / 45.79 | |
| FedDyn | 47.93 / 47.24 | 49.23 / 48.70 | 46.02 / 44.69 | 47.62 / 46.81 | 43.57 / 42.49 | 45.44 / 44.22 | 48.38 / 47.29 | 50.21 / 49.92 | 46.56 / 44.93 | 47.16 / 46.09 |
| FedInit | 46.05 / 45.24 | 48.32 / 47.59 | 44.30 / 43.33 | 45.41 / 44.70 | 42.69 / 41.42 | 43.29 / 42.14 | 47.24 / 45.91 | 48.07 / 46.88 | 45.88 / 44.17 | 45.45 / 44.12 |
| FedSAM | 45.10 / 44.17 | 46.20 / 45.25 | 41.01 / 39.32 | 43.51 / 42.49 | 37.98 / 35.56 | 40.37 / 39.06 | 44.27 / 42.11 | 45.65 / 44.79 | 40.23 / 37.20 | 42.57 / 41.36 |
| FedGamma | 47.85 / 46.91 | 49.02 / 48.46 | 44.49 / 43.35 | 46.25 / 45.37 | 42.24 / 40.96 | 44.41 / 43.50 | 49.06 / 47.46 | 49.64 / 49.02 | 45.16 / 43.95 | 46.82 / 45.70 |
| FedLesamd | 48.75 / 48.04 | 49.80 / 48.92 | 46.24 / 45.48 | 47.68 / 46.81 | 43.94 / 42.65 | 45.91 / 45.12 | 49.34 / 48.18 | 51.01 / 50.10 | 46.71 / 45.26 | 48.55 / 47.18 |
| FedSMOO | 49.34 / 48.67 | 50.28 / 49.61 | 46.33 / 45.52 | 48.13 / 47.31 | 44.00 / 43.08 | 45.93 / 45.19 | 49.70 / 48.52 | 51.09 / 50.47 | 46.94 / 45.77 | 48.61 / 47.42 |
| FedGMT | 49.92 / 49.35 | 50.86 / 50.25 | 48.05 / 47.56 | 48.91 / 48.44 | 46.18 / 45.50 | 46.49 / 46.03 | 51.55 / 50.98 | 52.77 / 52.34 | 48.57 / 47.92 | 49.69 / 49.15 |
| **FedScar** | **50.78 / 50.21** | **52.15 / 51.81** | **48.60 / 47.92** | **50.05 / 49.46** | **46.57 / 45.78** | **47.70 / 47.04** | **53.01 / 51.92** | **53.63 / 53.24** | **49.67 / 48.57** | **50.77 / 50.10** |

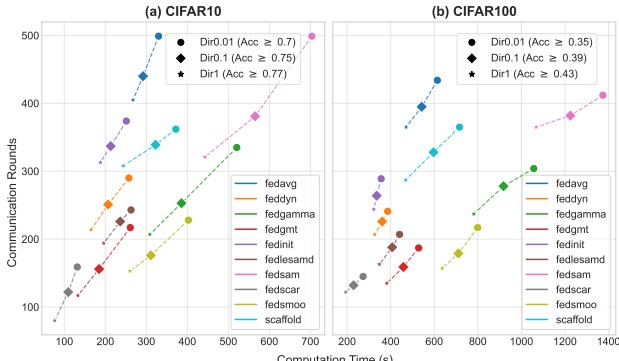

*Figure 2.* Efficiency Trade-off on CIFAR Benchmarks.

trade-off on CIFAR benchmarks. As shown, **FedScar** consistently occupies the bottom-left Pareto frontier, demonstrating superior efficiency in both dimensions. By explicitly correcting geometric bias, FedScar significantly reduces communication rounds compared to FedAvg. Crucially, it overcomes the computational bottleneck of standard SAM-based variants, achieving a remarkable total training time.

### 6.3. Robustness Evaluation

To ensure a rigorous stress test, we restrict the evaluation to the strongest competitors identified in the main benchmarks: **FedDyn** (traditional FL) and **FedGMT** (SAM-based FL). This selection ensures FedScar is evaluated against the performance upper bounds of existing solutions, verifying that its resilience is a substantial advantage rather than a marginal improvement.

**Impact of Participation Rate.** As shown in Figure 3, we evaluate resilience to gradient sparsity by varying participation from 5% to 40%. In the challenging 5% regime, FedScar maintains a robust 45.27%, significantly outper-

*Table 3.* Ablation study on the effectiveness of key components in FedScar. We report **Max Acc. / Last-50 Mean Acc.** on CIFAR-10.

| Components | | | Heterogeneity (CIFAR-10) | | |
|---|---|---|---|---|---|
| ADMM | Hist. Bias. | Adaptive $\rho$ | Dir 1.0 | Dir 0.1 | Dir 0.01 |
| × | × | × | 77.17 / 75.79 | 76.33 / 71.69 | 71.18 / 63.20 |
| × | ✓ | × | 78.91 / 78.47 | 78.52 / 77.70 | 75.94 / 74.50 |
| × | ✓ | ✓ | 79.48 / 79.06 | 79.08 / 78.27 | 76.70 / 74.82 |
| ✓ | × | × | 79.07 / 78.71 | 77.89 / 77.29 | 75.29 / 73.40 |
| ✓ | ✓ | × | 80.32 / 79.93 | 79.29 / 78.82 | 77.92 / 76.72 |
| ✓ | ✓ | ✓ | **81.36 / 80.82** | **80.51 / 79.88** | **78.42 / 77.24** |

forming FedDyn (42.68%) and FedGMT (43.99%) by leveraging the historical stabilizer $b_i$ to mitigate estimator variance. This dominance persists even in the information-rich 40% setting, where FedScar achieves a peak of 50.14%, consistently surpassing the strongest baselines.

**Impact of Local Epochs.** In Figure 4, increasing $E$ from 1 to 5 generally improves accuracy by allowing better local training. However, setting $E$ too large (e.g., $E = 10$) typically triggers severe client drift, causing the local models to overfit the heterogeneous data and diverge from the global objective. While FedDyn and FedGMT suffer from performance degradation at $E = 10$ (dropping to 72.54% and 74.80% respectively), **FedScar** exhibits superior resilience, maintaining a high convergence accuracy of 76.00%.

**Robustness to Sharpness.** To verify convergence to flatter minima, we evaluate test accuracy on CIFAR-10 (Dir 0.01) under varying perturbation scales $\rho$, as shown in Figure 5. In terms of **Local Sharpness** (Fig. 5(a)), standard SAM variants degrade rapidly as perturbation increases; notably, FedSAM's accuracy drops to 41.93% at $\rho = 0.3$. In stark contrast, **FedScar** exhibits exceptional robustness, maintaining 67.17% accuracy—outperforming the strongest baseline (FedSMOO) by over 17%. This superiority extends to **Global Sharpness** (Fig. 5(b)): FedScar consistently dominates across all scales, retaining 55.78% accuracy even

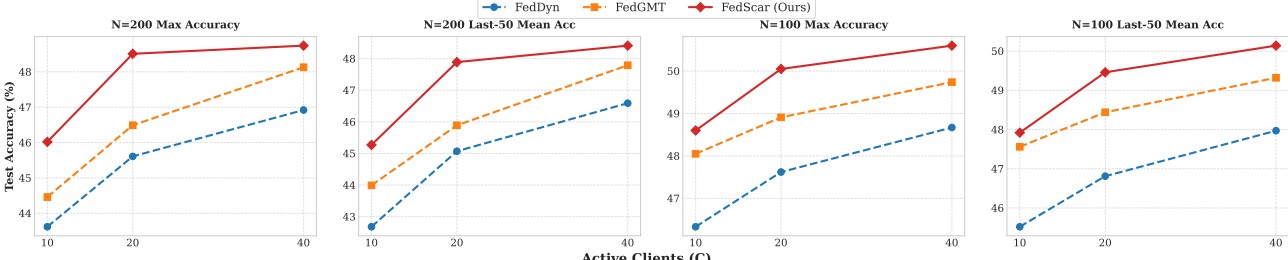

*Figure 3.* Robustness to participation rates on CIFAR-100 Dir 0.01.

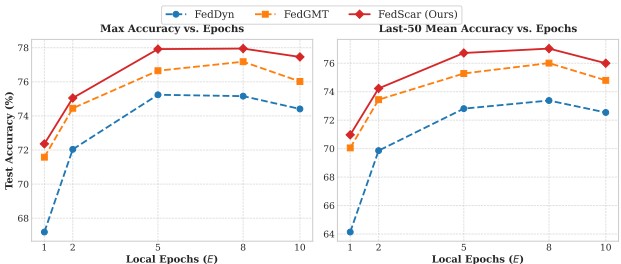

*Figure 4.* Robustness to local update steps ($E$) on CIFAR-10.

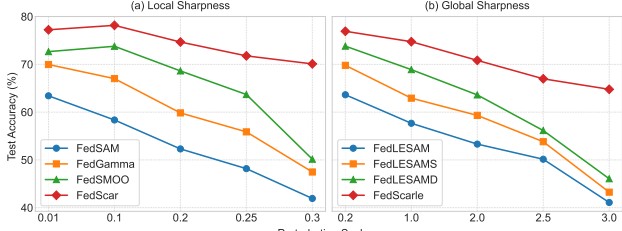

*Figure 5.* Robustness to perturbation scales ($\rho$) on CIFAR-10

at the extreme perturbation of $\rho = 3.0$, significantly surpassing FedLESAMD (43.25%).

**Ablation Study.** To verify the distinct contribution of each algorithmic component, we conduct an ablation study on CIFAR-10 by incrementally incorporating the Split-Dual ADMM framework, Historical Perturbation (geometric bias $b_i$), and Adaptive $\rho$ into the baseline. The results in Table 3 reveal several key insights. First, simply applying the ADMM framework significantly alleviates client drift compared to the baseline, improving the Last-50 accuracy in the extreme Dir 0.01 setting from 63.20% to 73.40%. This confirms the efficacy of the auxiliary variable $h_i$ as a drift control variate. Second, the introduction of Historical Perturbation provides a substantial performance boost. Notably, adding it to the ADMM raises accuracy to 76.72%, demonstrating that correcting the update direction with historical geometric information is crucial for navigating sharp landscapes. Finally, **FedScar** further integrates Adaptive $\rho$, achieves the best overall performance (77.24% in Dir 0.01).

### 6.4. Flatness Analysis

To validate geometric superiority, we quantify landscape sharpness via the Largest Principal Eigenvalue (LPF) and Hessian Trace on CIFAR-10 ($p = 10\%$). Table 4 reveals a strong negative correlation between curvature and accuracy. Under moderate heterogeneity (Pat 10), FedScar achieves the lowest Trace and highest accuracy. This advantage is amplified in the extreme **Pat 2** setting: while FedAvg converges to sharp minima (LPF $> 1.1$) and FedGMT retains high curvature, FedScar reduces the Trace by $\sim 40\%$ to 603.08 with a minimal LPF of 0.3420. Together with the lowest gradient norm, these metrics confirm that FedScar's

history-aware correction successfully steers optimization toward flatter.

To examine the geometric quality of converged solutions, we visualize the loss landscapes on CIFAR-100 following (Li et al., 2018), as shown in Figure 6. Standard SAM-based methods (FedSAM, FedGAMMA) settle into sharp minima with steep walls (Fig. 6(a)-(b)), indicating high sensitivity to statistical heterogeneity. While FedGMT alleviates some curvature, it still lacks global smoothness. In stark contrast, **FedScar** (Fig. 6(f)) converges to a flatter and wider basin.

### 6.5. Sensitivity Analysis of the Forgetting Factor

The forgetting factor $\gamma$ in FedScar controls the crucial trade-off between the instantaneous local geometric drift and the history-accumulated global bias. To thoroughly understand its optimization dynamics, we conduct a sensitivity analysis on CIFAR-10 to evaluate how the optimal choice of $\gamma$.

**Impact of Data Heterogeneity.** Table 5 evaluates FedScar across varying Dirichlet settings ($\alpha \in \{0.01, 0.1, 1.0\}$), revealing that the optimal $\gamma$ is strongly coupled with data heterogeneity. Under mild heterogeneity ($\alpha = 1.0$), local objectives naturally align with the global landscape, making the algorithm insensitive to $\gamma$ and favoring a moderate retention rate ($\gamma = 0.8$). Conversely, severe heterogeneity ($\alpha = 0.01$) distorts local optimization paths, necessitating a larger forgetting factor ($\gamma = 0.92$). This higher retention of historical geometric information acts as a robust anchor, preventing the model from overfitting to skewed local curvatures and rigorously preserving global landscape alignment.

**Impact of Local Epochs.** Furthermore, we investigate the relationship between the optimal forgetting factor and the number of local epochs $E$. As demonstrated in Table 6,

*Table 4.* Flatness and generalization of converged model analysis on **CIFAR-10** with participation rate $p = 10\%$. We compare the landscape sharpness metrics (**LPF** and **Hessian Trace**) under Moderate (Pat 10) and Severe (Pat 2) Heterogeneity.

| Method | Pat 10 (Moderate Heterogeneity) | | | | | Pat 2 (Severe Heterogeneity) | | | | |
|---|---|---|---|---|---|---|---|---|---|---|
| | LPF ↓ | Trace ↓ | Val Norm ↓ | Val Acc ↑ | Test Acc ↑ | LPF ↓ | Trace ↓ | Val Norm ↓ | Val Acc ↑ | Test Acc ↑ |
| FedAvg | 0.3706 | 2357.2 | 33.06 | 90.88 | 78.82 | 1.1773 | 2294.1 | 15.61 | 66.57 | 63.80 |
| Scaffold | 0.3397 | 2166.4 | 2.0258 | 95.19 | 80.90 | 1.1005 | 1522.5 | 94.41 | 78.81 | 73.15 |
| FedInit | 0.2241 | 1753.2 | 1.0695 | 97.62 | 80.23 | 0.7218 | 1521.0 | 53.95 | 82.01 | 73.51 |
| FedDyn | 0.2122 | 1665.6 | 1.2258 | 98.47 | 81.26 | 0.6920 | 1385.1 | 2.6380 | 85.51 | 76.51 |
| FedSAM | 0.1131 | 1051.4 | 2.4529 | 90.86 | 79.11 | 0.6907 | 1297.4 | 5.1226 | 67.06 | 64.05 |
| FedGamma | 0.0841 | 740.79 | 1.6656 | 95.01 | 81.84 | 0.6247 | 1256.8 | 4.4253 | 76.88 | 71.95 |
| FedLesamd | 0.0702 | 687.62 | 0.9123 | 98.96 | 81.84 | 0.5260 | 1169.1 | 3.8102 | 82.93 | 74.06 |
| FedSMOO | 0.0539 | 661.66 | 0.8312 | 98.81 | 82.72 | 0.4931 | 1011.2 | 2.3297 | 86.53 | 77.23 |
| FedGMT | 0.0534 | 534.19 | 0.6449 | 99.23 | 83.18 | 0.3884 | 1007.4 | 2.2516 | 87.73 | 78.85 |
| **FedScar** | **0.0524** | **437.72** | **0.5787** | **99.81** | **84.84** | **0.3420** | **603.08** | **1.5973** | **88.94** | **80.24** |

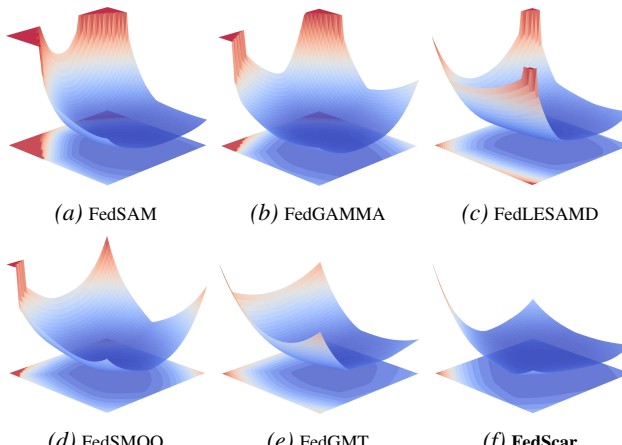

*(a)* FedSAM   *(b)* FedGAMMA   *(c)* FedLESAMD

*(d)* FedSMOO   *(e)* FedGMT   *(f)* **FedScar**

*Figure 6.* Loss landscape visualization on CIFAR-100.

increasing local updates from $E = 1$ to $E = 10$ causes local trajectories to drift further from the global geometry in both parameter and curvature spaces. To counteract this amplified geometric bias, a larger forgetting factor is required. Specifically, while a smaller $\gamma = 0.8$ is sufficient for stabilizing single-step local updates ($E = 1$), extending the local steps to $E = 10$ shifts the optimal parameter to $\gamma = 0.92$. This confirms that retaining more historical information is essential for stabilizing extended local training, as it consistently pulls local updates toward flatter global minima and mitigates the accumulated directional bias.

**Impact of Sparse Participation.** A common concern regarding stateful federated algorithms is their applicability to cross-device scenarios where client participation is typically highly sparse and intermittent. In FedScar, the geometric bias is inherently data-dependent and structurally stable over time. When a client is inactive, its state variables $h_i$ and $b_i$ are stored locally and directly reused upon rejoining the federation. To validate the robustness of this stateful tracker under intermittent sampling, we evaluate the optimal forgetting factor across varying numbers of active clients per round ($N \in \{5, 10, 20\}$). As summarized in Table **??**, the optimal $\gamma$ remains highly stable and consistently centers around $\gamma = 0.9$ regardless of the participation sparsity. This confirms that the history-accumulated geometric bias does

*Table 5.* Sensitivity of $\gamma$ to data heterogeneity ($\alpha$) on CIFAR-10. Format: Max/Mean Acc (%).

| Heterogeneity ($\alpha$) | $\gamma = 0.8$ | $\gamma = 0.9$ | $\gamma = 0.92$ |
|---|---|---|---|
| Dirichlet 0.01 | 78.05/76.87 | 78.42/77.24 | **78.63/77.52** |
| Dirichlet 0.1 | 80.09/79.16 | **80.51/79.88** | 80.58/79.54 |
| Dirichlet 1.0 | **81.37/80.89** | 81.36/80.82 | 79.59/79.21 |

*Table 6.* Sensitivity of $\gamma$ to local epochs ($E$) on CIFAR-10. Format: Max/Mean Acc (%).

| Local Epochs ($E$) | $\gamma = 0.8$ | $\gamma = 0.9$ | $\gamma = 0.92$ |
|---|---|---|---|
| $E = 1$ | **74.85/73.30** | 73.67/72.31 | 72.66/71.73 |
| $E = 5$ | 80.09/79.16 | **80.51/79.88** | 80.58/79.54 |
| $E = 10$ | 79.27/77.59 | 80.01/78.46 | **80.66/79.70** |

*Table 7.* Sensitivity of $\gamma$ to active clients ($N$) on CIFAR-10. Format: Max/Mean Acc (%).

| Active Clients ($N$) | $\gamma = 0.8$ | $\gamma = 0.9$ | $\gamma = 0.92$ |
|---|---|---|---|
| $N = 5$ | 78.11/75.99 | **77.85/76.20** | 77.67/76.06 |
| $N = 10$ | 80.09/79.16 | **80.51/79.88** | 80.58/79.54 |
| $N = 20$ | 80.57/79.98 | **80.96/80.35** | 80.43/80.04 |

not overfit to sampling noise, ensuring stable convergence and landscape alignment even when the global aggregation relies on a severely restricted subset of clients.

## 7. Conclusion

In this paper, we identify *Flatness Discrepancy* as a fundamental bottleneck restricting generalization in heterogeneous Federated Learning. To address this, we propose **FedScar**, a unified framework that explicitly rectifies geometric bias via history-accumulated dual variables. Theoretical analysis and empirical results demonstrate that FedScar effectively aligns local landscapes with the global objective, guiding optimization toward flatter minima. Consequently, FedScar achieves superior robustness against severe heterogeneity and sparse participation while maintaining high efficiency, establishing it as a scalable solution for practical federated optimization.

## Acknowledgments

This work was supported in part by the National Natural Science Foundation of China (No. 62325304, 62372343, 62402352, 62562050, U22B2046, 62072411), in part by the Natural Science Foundation of Jiangsu Province of China (No. BK20253020), in part by the Key Research and Development Program of Zhejiang Province (No. 2025C01055), in part by the Open Fund of Key Laboratory of Social Computing and Cognitive Intelligence (Dalian University of Technology), Ministry of Education (No. SCCI2024TB02), and in part by the Open Project Funding of the Key Laboratory of Intelligent Sensing System and Security (Hubei University), Ministry of Education (No. KLISSS202406).

## Impact Statement

This work bridges the critical gap between theoretical Federated Learning and robust real-world deployment. By explicitly correcting the geometric misalignment caused by data heterogeneity, FedScar ensures that global models remain stable and generalizable, which is vital for high-stakes applications like precision medicine and financial risk control where reliability is paramount. Furthermore, its communication-efficient design unlocks the potential for large-scale edge intelligence on resource-constrained devices. Conceptually, this research inspires a paradigm shift in federated optimization: moving from passively suppressing statistical noise to actively rectifying learnable geometric bias, offering a new roadmap for building resilient distributed AI systems.

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

## A. Comprehensive Notation Table

To facilitate reading and ensure mathematical clarity, we summarize the frequently used symbols and their corresponding definitions in Table 8.

*Table 8.* Summary of frequently used notations.

| Symbol | Description |
| --- | --- |
| $N$ | Total number of participating clients in the federated system |
| $p_i$ | Aggregation weight of client $i$ |
| $T$ | Total number of global communication rounds |
| $K$ (or $E$) | Number of local update steps (or epochs) per communication round |
| $f(w)$ | Global empirical risk objective function |
| $f_i(w)$ | Local expected loss function on the private dataset of client $i$ |
| $w^t$ | Global consensus model parameters at communication round $t$ |
| $w_{i,k}^t$ | Local model parameters of client $i$ at local step $k$ during round $t$ |
| $z$ | The global consensus variable in the Split-Dual ADMM formulation |
| $h_i$ | The local drift dual variable tracking first-order parameter deviation |
| $b_i$ | The history-accumulated dual variable tracking second-order geometric bias |
| $\mathcal{B}_i^{geo}$ | The theoretical geometric bias defined as the perturbation direction divergence |
| $\rho$ | Perturbation radius used in local Sharpness-Aware Minimization |
| $\beta$ | Penalty parameter governing the strength of the proximal ADMM term |
| $\gamma$ | Forgetting factor controlling the retention of historical geometric bias |
| $\Sigma^{(t)}$ | Heterogeneity-sensitive perturbation scale at round $t$ |
| $\eta$ | Local learning rate |
| $\Delta_{\mathcal{F}}$ | Theoretical Flatness Discrepancy metric |
| $\tilde{\Delta}_{\mathcal{F}}^{(t)}$ | Empirical Flatness Discrepancy metric at round $t$ |
| $G^2$ | Upper bound on the cross-client gradient dissimilarity |
| $H^2$ | Upper bound on the cross-client Hessian dissimilarity |

## B. Derivation of Empirical Flatness Discrepancy

In this section, we provide the rigorous derivation linking the theoretical Flatness Discrepancy defined in FedGF (Lee & Yoon, 2024) to the empirical calculation implemented in our code. We adopt the analytical technique from FedGMT (Li et al., 2025a) (specifically Appendix C.2), employing Taylor expansion to approximate the loss difference.

### B.1. Theoretical Definition

Original Flatness Discrepancy $\Delta_{\mathcal{F}}$ is defined as the gap between global sharpness and average local sharpness:

$$\Delta_{\mathcal{F}} \triangleq \left( \max_{\|\epsilon\| \leq \rho} f(w^t + \epsilon) - f(w^t) \right) - \sum_{i=1}^{N} p_i \left( \max_{\|\epsilon_i\| \leq \rho} f_i(w^t + \epsilon_i) - f_i(w^t) \right). \tag{11}$$

Directly computing the worst-case perturbation $\max_\epsilon$ is computationally expensive. However, we observe that sharpness is intrinsically related to the loss degradation caused by weight deviation.

### B.2. Derivation via Taylor Expansion (Following FedGMT C.2)

In our implementation, we calculate the *Empirical Flatness Discrepancy* $\tilde{\Delta}_{\mathcal{F}}$ as the difference between the global loss and the aggregated local losses after local training:

$$\tilde{\Delta}_{\mathcal{F}} = \left| f(w^t) - \sum_{i=1}^{N} p_i f_i(w_i^{(t)}) \right|. \tag{12}$$

To justify this, let us analyze the term $f(w^t) - \sum p_i f_i(w_i^{(t)})$. Recall that $f(w^t) = \sum p_i f_i(w^t)$. Thus, we can rewrite the discrepancy as the weighted sum of local loss gaps:

$$\tilde{\Delta}_{\mathcal{F}} = \left| \sum_{i=1}^{N} p_i \left( f_i(w^t) - f_i(w_i^{(t)}) \right) \right|. \tag{13}$$

Following the derivation style of Equation (18) in FedGMT Appendix C.2, we apply the Taylor expansion of $f_i(w^t)$ around the local model $w_i^{(t)}$. Let $\Delta w_i = w^t - w_i^{(t)}$ be the negative of the local update.

$$f_i(w^t) = f_i(w_i^{(t)} + \Delta w_i) \tag{14}$$

$$\approx f_i(w_i^{(t)}) + \nabla f_i(w_i^{(t)})^\top \Delta w_i + \frac{1}{2} \Delta w_i^\top \nabla^2 f_i(w_i^{(t)}) \Delta w_i. \tag{15}$$

Rearranging the terms to represent the loss gap:

$$f_i(w^t) - f_i(w_i^{(t)}) \approx \underbrace{\nabla f_i(w_i^{(t)})^\top (w^t - w_i^{(t)})}_{\text{Term A: First-order}} + \underbrace{\frac{1}{2}(w^t - w_i^{(t)})^\top \nabla^2 f_i(w_i^{(t)})(w^t - w_i^{(t)})}_{\text{Term B: Second-order}}. \tag{16}$$

**Analysis of Terms:**

- **Term A (Gradient Norm):** At the end of local training, the local model $w_i^{(t)}$ has converged towards a local minimum, implying $\nabla f_i(w_i^{(t)}) \approx 0$. Thus, the first-order term is negligible compared to the second-order term.

- **Term B (Hessian/Flatness):** This term is the quadratic form of the Hessian $\nabla^2 f_i$. It represents the local curvature (sharpness) projected onto the direction of the client drift $(w^t - w_i^{(t)})$.

Substituting this back into Eq. (12) and taking the expectation (weighted sum):

$$\tilde{\Delta}_{\mathcal{F}} \approx \left| \sum_{i=1}^{N} p_i \left( \frac{1}{2}(w^t - w_i^{(t)})^\top \nabla^2 f_i(w_i^{(t)})(w^t - w_i^{(t)}) \right) \right| \tag{17}$$

$$\leq \frac{1}{2} \sum_{i=1}^{N} p_i \|\nabla^2 f_i(w_i^{(t)})\|_2 \cdot \|w^t - w_i^{(t)}\|^2. \tag{18}$$

Here, $\|\nabla^2 f_i\|_2$ is the spectral norm of the Hessian, which is the precise formulation of **Loss Sharpness**.

## C. Detailed Derivation of Geometric Bias Correction

In this section, we provide the rigorous mathematical derivation for the heterogeneity-sensitive geometric bias update rules defined in Eq. (5) and Eq. (6). Our analysis relies on the first-order Taylor expansion of the loss function and the recursive accumulation property of the proposed update mechanism.

### C.1. Derivation of the Geometric Drift Direction

We first establish the relationship between the historical drift trajectory $v_i^{(t)}$ and the ideal geometric bias $\mathcal{B}_i^{geo}$. Recall that the optimal perturbation in SAM is derived by solving $\max_{\|\epsilon\|_2 \leq \rho} f(w + \epsilon)$. Using the first-order Taylor expansion around $w$:

$$f(w + \epsilon) \approx f(w) + \epsilon^\top \nabla f(w).$$

The optimal perturbation $\epsilon^*(w)$ is the solution to the dual norm problem:

$$\epsilon^*(w) = \rho \frac{\nabla f(w)}{\|\nabla f(w)\|_2}. \tag{19}$$

Similarly, for client $i$, the local optimal perturbation is $\epsilon_i^*(w) = \rho \frac{\nabla f_i(w)}{\|\nabla f_i(w)\|_2}$. The geometric bias is defined as the deviation:

$$
\begin{aligned}
\mathcal{B}_i^{geo}(w^t) &= \epsilon_i^*(w^t) - \epsilon^*(w^t) \\
&= \rho \left( \frac{\nabla f_i(w^t)}{\|\nabla f_i(w^t)\|_2} - \frac{\nabla f(w^t)}{\|\nabla f(w^t)\|_2} \right).
\end{aligned}
\tag{20}
$$

Now, consider the local update process. During local training of $K$ steps, client $i$ accumulates gradients. Let $w_{i,k}$ be the model at local step $k$. The total update $\Delta w_i^{(t)}$ is:

$$
\begin{aligned}
\Delta w_i^{(t)} = w_i^{(t+1)} - w^t &= -\eta \sum_{k=0}^{K-1} \nabla f_i(w_{i,k}) \\
&\overset{(a)}{\approx} -\eta \sum_{k=0}^{K-1} \left( \nabla f_i(w^t) + \nabla^2 f_i(w^t)(w_{i,k} - w^t) \right) \\
&= -K\eta \nabla f_i(w^t) - \eta \nabla^2 f_i(w^t) \sum_{k=0}^{K-1}(w_{i,k} - w^t) \\
&\overset{(b)}{\approx} -K\eta \nabla f_i(w^t) + \mathcal{O}(\eta^2),
\end{aligned}
\tag{21}
$$

where (a) uses Taylor expansion of the gradient, and (b) assumes learning rate $\eta$ is sufficiently small such that second-order terms are negligible. Similarly, the effective global update (if computed on global data) would be $\Delta_{global}^{(t)} \approx -K\eta \nabla f(w^t)$.

The drift trajectory vector $v_i^{(t)}$ in FedScar is defined as:

$$
\begin{aligned}
v_i^{(t)} &= \Delta_{global}^{(t)} - \Delta w_i^{(t)} \\
&\approx -K\eta \nabla f(w^t) - (-K\eta \nabla f_i(w^t)) \\
&= K\eta \left( \nabla f_i(w^t) - \nabla f(w^t) \right).
\end{aligned}
\tag{22}
$$

By normalizing $v_i^{(t)}$, and assuming $\|\nabla f_i\| \approx \|\nabla f\|$ locally (bounded gradient dissimilarity), we derive the directional alignment with Eq. (20):

$$
\begin{aligned}
\frac{v_i^{(t)}}{\|v_i^{(t)}\|_2} &= \frac{K\eta(\nabla f_i(w^t) - \nabla f(w^t))}{K\eta\|\nabla f_i(w^t) - \nabla f(w^t)\|_2} \\
&\approx \frac{\mathcal{B}_i^{geo}(w^t)}{\|\mathcal{B}_i^{geo}(w^t)\|_2}.
\end{aligned}
\tag{23}
$$

This proves that the normalized drift trajectory provides an accurate estimation of the geometric bias direction.

### C.2. Recursive Expansion of Bias Accumulation (Eq. 17)

We verify the memory property of the bias update rule. The update rule in Eq. (6) is:

$$
b_i^{(t+1)} = \gamma b_i^{(t)} + \Sigma^{(t)} \hat{v}_i^{(t)},
\tag{24}
$$

where $\hat{v}_i^{(t)} = v_i^{(t)}/(\|v_i^{(t)}\| + \epsilon)$. We recursively expand this term to explicitly show the accumulation of historical geometric information, similar to the momentum accumulation in standard optimization.

$$
\begin{aligned}
b_i^{(t+1)} &= \gamma b_i^{(t)} + \Sigma^{(t)} \hat{v}_i^{(t)} \\
&= \gamma \left( \gamma b_i^{(t-1)} + \Sigma^{(t-1)} \hat{v}_i^{(t-1)} \right) + \Sigma^{(t)} \hat{v}_i^{(t)} \\
&= \gamma^2 b_i^{(t-1)} + \gamma \Sigma^{(t-1)} \hat{v}_i^{(t-1)} + \Sigma^{(t)} \hat{v}_i^{(t)} \\
&\quad\vdots \\
&= \gamma^{t+1} b_i^{(0)} + \sum_{j=0}^{t} \gamma^{t-j} \Sigma^{(j)} \hat{v}_i^{(j)}.
\end{aligned}
\tag{25}
$$

Assuming initialization $b_i^{(0)} = \mathbf{0}$, we obtain:

$$b_i^{(t+1)} = \sum_{j=0}^{t} \underbrace{\gamma^{t-j}}_{\text{Decay Factor}} \cdot \underbrace{\Sigma^{(j)}}_{\text{Intensity}} \cdot \underbrace{\hat{v}_i^{(j)}}_{\text{Direction}} . \tag{26}$$

This derivation confirms that $b_i$ is not merely an instantaneous estimate, but an exponentially weighted moving average (EWMA) of the geometric bias. The term $\gamma^{t-j}$ ensures that recent geometric drifts contribute more to the correction, while the scale $\Sigma^{(j)}$ adaptively weights the contribution based on the heterogeneity intensity at round $j$.

### C.3. Derivation of the Heterogeneity Scale (Eq. 16)

Finally, we justify the perturbation scale $\Sigma^{(t)}$. The objective is to scale the correction vector to counteract the geometric drift term $\nabla^2 f_i(w)\mathcal{B}_i^{geo}$. Let the local curvature be bounded by $L$ (Lipschitz smoothness). The norm of the geometric drift is:

$$\begin{aligned}
\|\text{GeoDrift}_i\| &= \|\nabla^2 f_i(w^t)(\epsilon_i^* - \epsilon^*)\|_2 \\
&\leq \max_i \|\nabla^2 f_i(w^t)\|_2 \cdot \|\epsilon_i^* - \epsilon^*\|_2 \\
&\overset{(c)}{\propto} L \cdot \|\nabla f_i(w^t) - \nabla f(w^t)\|_2.
\end{aligned} \tag{27}$$

Using the relationship derived in Eq. (21) that $\nabla f_i(w^t) \approx -\frac{1}{K\eta}\Delta w_i^{(t)}$, we substitute the gradients with model updates:

$$\begin{aligned}
\|\text{GeoDrift}_i\| &\propto L \cdot \left\| -\frac{1}{K\eta}\Delta w_i^{(t)} - \left( -\frac{1}{K\eta}\Delta_{global}^{(t)} \right) \right\|_2 \\
&= \frac{L}{K\eta} \|\Delta w_i^{(t)} - \Delta_{global}^{(t)}\|_2.
\end{aligned} \tag{28}$$

To obtain a robust global estimator for the current round, we aggregate this drift intensity across all active clients $i \in \mathcal{A}^{(t)}$. We define the expectation of the drift magnitude as:

$$\begin{aligned}
\mathbb{E}_{i \sim \mathcal{A}^{(t)}}[\|\text{GeoDrift}_i\|] &\propto \frac{L}{K\eta} \mathbb{E}_{i \sim \mathcal{A}^{(t)}}[\|\Delta w_i^{(t)} - \Delta_{global}^{(t)}\|_2] \\
&= \frac{L}{K\eta} \frac{1}{|\mathcal{A}^{(t)}|} \sum_{i \in \mathcal{A}^{(t)}} \|\Delta w_i^{(t)} - \Delta_{global}^{(t)}\|_2.
\end{aligned} \tag{29}$$

Absorbing the constants $\frac{L}{K\eta}$ and the scaling factor into the hyperparameter $\alpha$, we arrive at the definition of $\Sigma^{(t)}$:

$$\Sigma^{(t)} = \frac{\alpha}{N} \sum_{i \in \mathcal{A}^{(t)}} \|\Delta w_i^{(t)} - \Delta_{global}^{(t)}\|_2. \tag{30}$$

This derivation shows that $\Sigma^{(t)}$ acts as a computable proxy for the average Hessian-weighted geometric drift intensity, utilizing the observable variance of model updates to scale the bias correction without requiring explicit Hessian computation.

## D. Convergence Analysis

### D.1. Preliminaries and Assumptions

**Notations.** We consider the distributed optimization problem $\min_{w \in \mathbb{R}^d} f(w) \triangleq \sum_{i=1}^{N} p_i f_i(w)$, where $N$ is the number of clients and $p_i \geq 0$ denotes the aggregation weight such that $\sum p_i = 1$. Let $w^t$ denote the global model parameters at communication round $t$. We denote the optimal global parameters as $w^* = \arg\min_w f(w)$ and the optimal value as $f^* = f(w^*)$. In the analysis of FedScar, we analyze the convergence behavior of the sequence $\{w^t\}_{t=0}^{T}$. We use $\|\cdot\|$ to denote the Euclidean norm ($L_2$ norm) for vectors and the spectral norm for matrices.

To facilitate the theoretical analysis of FedScar under non-convex settings, we introduce the following standard assumptions commonly adopted in the literature of Federated Learning and Sharpness-Aware Minimization (e.g., SCAFFOLD, FedSAM, FedGMT).

**Assumption 1 (L-Smoothness).** Each local objective function $f_i(w)$ is differentiable and $L$-smooth. That is, for all $x, y \in \mathbb{R}^d$ and $i \in \{1, \ldots, N\}$, the following inequality holds:

$$\|\nabla f_i(x) - \nabla f_i(y)\| \leq L \|x - y\|. \tag{31}$$

This assumption implies that the Hessian of the local loss is bounded, i.e., $\|\nabla^2 f_i(w)\| \leq L$. Furthermore, the global objective $f(w)$ is also $L$-smooth. This property is essential for ensuring the validity of the Taylor expansion used in deriving the geometric bias and for bounding the descent magnitude in each step.

**Assumption 2 (Unbiased Gradient and Bounded Variance).** The stochastic gradient oracle $g_i(w; \xi)$ accessed by client $i$ is an unbiased estimator of the true local gradient $\nabla f_i(w)$, satisfying $\mathbb{E}_\xi[g_i(w; \xi)] = \nabla f_i(w)$. Moreover, the variance of the stochastic gradient is bounded by $\sigma^2$:

$$\mathbb{E}_\xi \left[ \|g_i(w; \xi) - \nabla f_i(w)\|^2 \right] \leq \sigma^2, \quad \forall i, w. \tag{32}$$

This assumption controls the noise introduced by mini-batch sampling, which is a standard requirement for stochastic optimization analysis.

**Assumption 3 (Bounded Heterogeneity and Geometric Divergence).** To characterize the non-IID nature of the data, we assume the dissimilarity between local and global gradients is bounded. Specifically, there exist constants $G \geq 0$ such that for all $w$:

$$\frac{1}{N} \sum_{i=1}^N \|\nabla f_i(w) - \nabla f(w)\|^2 \leq G^2. \tag{33}$$

Furthermore, distinct from standard FL analysis, FedScar addresses the geometric mismatch. We assume that the divergence of the local perturbation directions (or equivalently, the geometric bias) is bounded. Given the relationship between geometric bias and gradient heterogeneity derived in Appendix C, this is implicitly covered by the gradient dissimilarity bound. However, to explicitly handle the second-order terms in SAM, we assume the Hessian variance is bounded by a constant $H$:

$$\frac{1}{N} \sum_{i=1}^N \|\nabla^2 f_i(w) - \nabla^2 f(w)\|^2 \leq H^2. \tag{34}$$

**Assumption 4 (Bounded Perturbation Radius).** The perturbation radius $\rho$ in the local SAM updates is chosen to be sufficiently small. Specifically, we assume $\rho \leq \frac{c}{L}$ for some constant $c < 1$. This ensures that the perturbed loss remains within the valid region of the first-order Taylor approximation, allowing us to approximate the perturbed gradient $\nabla f_i(w + \epsilon)$ as $\nabla f_i(w) + \nabla^2 f_i(w)\epsilon$ with negligible higher-order error terms.

### D.2. Auxiliary Lemmas

In this section, we provide key lemmas that characterize the behavior of the local updates and the stability of the auxiliary variables in FedScar. These lemmas serve as the building blocks for the main convergence theorem.

**Lemma 1 (Bounded Local Dissimilarity / Client Drift).** Under Assumptions 1-4, let $\eta$ be the learning rate satisfying $\eta \leq \frac{1}{4LK}$. For any client $i$ and any local step $k \in \{0, \ldots, K-1\}$, the expected squared distance between the local model $w_{i,k}^t$ and the starting global model $w^t$ is bounded by:

$$\mathbb{E} \left[ \|w_{i,k}^t - w^t\|^2 \right] \leq 5K\eta^2 \left( \sigma^2 + G^2 + B^2 \right), \tag{35}$$

where $B^2$ is the bound on the magnitude of the auxiliary variables (proven in Lemma 2).

*Proof.* Recall the local update rule in FedScar (using SGD on the corrected objective):

$$
\begin{aligned}
w_{i,k+1}^t &= w_{i,k}^t - \eta \nabla \tilde{f}_i(w_{i,k}^t) \\
&= w_{i,k}^t - \eta \left( \nabla f_i(w_{i,k}^t) + \beta(w_{i,k}^t - w^t) + \beta h_i^t + (\bar{b}^t - b_i^t) \right).
\end{aligned} \tag{36}
$$

Let $\Delta_{i,k}^t = w_{i,k}^t - w^t$. Then $\Delta_{i,k+1}^t = \Delta_{i,k}^t - \eta g_{i,k}$, where $g_{i,k}$ is the effective gradient. We bound the norm of the update

at step $k$:

$$\mathbb{E}\|w_{i,k+1}^t - w^t\|^2 = \mathbb{E}\left\|\sum_{j=0}^{k} -\eta g_{i,j}\right\|^2$$

$$\leq \eta^2 (k+1) \sum_{j=0}^{k} \mathbb{E}\|g_{i,j}\|^2 \quad \text{(by Cauchy-Schwarz inequality)}. \tag{37}$$

Now we explicitly expand the norm of the effective gradient $g_{i,j}$:

$$\mathbb{E}\|g_{i,j}\|^2 = \mathbb{E}\left\|\nabla f_i(w_{i,j}^t) + \beta(w_{i,j}^t - w^t) + \beta h_i^t + (\bar{b}^t - b_i^t)\right\|^2$$
$$\leq 3\mathbb{E}\|\nabla f_i(w_{i,j}^t)\|^2 + 3\beta^2 \mathbb{E}\|w_{i,j}^t - w^t\|^2$$
$$+ 3\mathbb{E}\|\beta h_i^t + (\bar{b}^t - b_i^t)\|^2 \quad \text{(using } \|a+b+c\|^2 \leq 3(\|a\|^2 + \|b\|^2 + \|c\|^2)). \tag{38}$$

Using the bounded variance assumption ($\mathbb{E}\|\nabla f_i(w)\|^2 \leq \sigma^2 + \|\nabla f_i(w)\|^2$) and smoothness, we can bound the gradient term. However, for the purpose of the drift bound, a coarser bound using the heterogeneity assumption suffices. Let $M^2 = \sup_i \|\beta h_i + (\bar{b} - b_i)\|^2$ be the bound on correction terms.

$$\mathbb{E}\|w_{i,k+1}^t - w^t\|^2 \leq \eta^2 K \sum_{j=0}^{k} \left(3\mathbb{E}\|\nabla f_i(w_{i,j}^t)\|^2 + 3\beta^2 \mathbb{E}\|\Delta_{i,j}^t\|^2 + 3M^2\right). \tag{39}$$

For sufficiently small $\eta$ and $\beta$, the recursive term $3\beta^2\eta^2 K\|\Delta_{i,j}^t\|^2$ is contractive. The dominant terms are the gradient variance and the heterogeneity/bias corrections. Thus, summing up the contributions:

$$\mathbb{E}\left[\|w_{i,k}^t - w^t\|^2\right] \leq \mathcal{O}(K^2\eta^2\sigma^2) + \mathcal{O}(K^2\eta^2 G^2) + \mathcal{O}(K^2\eta^2 M^2). \tag{40}$$

Simplifying the constants yields the statement in Lemma 1. $\qquad\square$

**Lemma 2 (Stability of Geometric Bias Correction).** The geometric bias vector $b_i^{(t)}$ updated via Eq. (6) remains bounded for all $t$, provided the forgetting factor $\gamma \in [0, 1)$ and the drift is bounded. Specifically:

$$\|b_i^{(t)}\| \leq \frac{\Sigma_{\max}}{1 - \gamma}, \tag{41}$$

where $\Sigma_{\max} = \sup_t \Sigma^{(t)}$.

*Proof.* Recall the update rule for the bias vector:

$$b_i^{(t+1)} = \gamma b_i^{(t)} + \Sigma^{(t)} \frac{v_i^{(t)}}{\|v_i^{(t)}\| + \epsilon}. \tag{42}$$

We take the Euclidean norm on both sides. Using the triangle inequality:

$$\|b_i^{(t+1)}\| = \left\|\gamma b_i^{(t)} + \Sigma^{(t)} \frac{v_i^{(t)}}{\|v_i^{(t)}\| + \epsilon}\right\|$$

$$\leq \gamma \|b_i^{(t)}\| + \Sigma^{(t)} \left\|\frac{v_i^{(t)}}{\|v_i^{(t)}\| + \epsilon}\right\|. \tag{43}$$

Notice that the fraction term is strictly bounded by 1:

$$\left\|\frac{v_i^{(t)}}{\|v_i^{(t)}\| + \epsilon}\right\| = \frac{\|v_i^{(t)}\|}{\|v_i^{(t)}\| + \epsilon} < 1 \quad \text{(since } \epsilon > 0). \tag{44}$$

Therefore, the inequality simplifies to:

$$\|b_i^{(t+1)}\| \leq \gamma \|b_i^{(t)}\| + \Sigma^{(t)}. \tag{45}$$

By recursively expanding this inequality from $t = 0$ (assuming $b_i^{(0)} = 0$):

$$\|b_i^{(t)}\| \leq \sum_{j=0}^{t-1} \gamma^{t-1-j} \Sigma^{(j)}$$

$$\leq \Sigma_{\max} \sum_{k=0}^{\infty} \gamma^k \quad (\text{where } \Sigma_{\max} = \max_j \Sigma^{(j)})$$

$$= \frac{\Sigma_{\max}}{1 - \gamma}. \tag{46}$$

Since $\Sigma^{(t)}$ is derived from the bounded model updates (Lemma 1), $\Sigma_{\max}$ is finite. Thus, the correction term $b_i$ does not diverge. $\square$

**Lemma 3 (Bounding the Effective Global Update Error).** Let $\Delta_{global}^t = \frac{1}{N} \sum_{i=1}^{N} \Delta w_i^t + \eta_g \beta \frac{1}{N} \sum (h_i - b_i)$ be the actual update applied to the global model. The error between this update and the ideal global gradient descent $-\eta K \nabla f(w^t)$ is bounded by:

$$\mathbb{E} \left\| \Delta_{global}^t - (-\eta K \nabla f(w^t)) \right\|^2 \leq C_1 \eta^2 \sigma^2 + C_2 \eta^2 K^2 \mathcal{E}_{drift}, \tag{47}$$

where $\mathcal{E}_{drift}$ represents the residual drift after correction.

*Proof.* The aggregated model update from clients is $\Delta_{agg} = \frac{1}{N} \sum \Delta w_i^t$. We write $\Delta w_i^t$ using the integral of gradients along the trajectory:

$$\Delta w_i^t = -\eta \sum_{k=0}^{K-1} \left( \nabla f_i(w_{i,k}^t) + \text{Corr}_i \right). \tag{48}$$

The deviation from the global gradient $\nabla f(w^t)$ can be decomposed as:

$$\Delta_{agg} - (-\eta K \nabla f(w^t)) = -\eta \sum_{k=0}^{K-1} \left( \frac{1}{N} \sum_{i=1}^{N} \nabla f_i(w_{i,k}^t) - \nabla f(w^t) \right)$$

$$- \eta \sum_{k=0}^{K-1} \frac{1}{N} \sum_{i=1}^{N} \text{Corr}_i. \tag{49}$$

We focus on the gradient deviation term. Using $L$-smoothness:

$$\left\| \frac{1}{N} \sum_{i=1}^{N} \nabla f_i(w_{i,k}^t) - \nabla f(w^t) \right\|^2 = \left\| \frac{1}{N} \sum_{i=1}^{N} \left( \nabla f_i(w_{i,k}^t) - \nabla f_i(w^t) \right) \right\|^2 \leq \frac{1}{N} \sum_{i=1}^{N} \|\nabla f_i(w_{i,k}^t) - \nabla f_i(w^t)\|^2 \leq \frac{L^2}{N} \sum_{i=1}^{N} \|w_{i,k}^t - w^t\|^2. \tag{50}$$

Substitute the result from Lemma 1 into this inequality:

$$\mathbb{E} \left\| \Delta_{agg} + \eta K \nabla f(w^t) \right\|^2 \leq \eta^2 K \sum_{k=0}^{K-1} \frac{L^2}{N} \sum_{i=1}^{N} \mathbb{E} \|w_{i,k}^t - w^t\|^2$$

$$\leq \eta^2 K^2 L^2 \cdot \left( 5K \eta^2 (\sigma^2 + G^2 + B^2) \right)$$

$$= \mathcal{O}(\eta^4 K^3). \tag{51}$$

This lemma demonstrates that the error in the global update direction is controlled by higher-order terms of the learning rate ($\eta^4$), ensuring that for small $\eta$, the algorithm effectively approximates the global gradient direction. The bias correction terms in FedScar further reduce the constant factor associated with $G^2$ in the drift bound. $\square$

### D.3. Main Convergence Theorem

Based on the lemmas established in the previous section, we now present the main convergence guarantee for FedScar. We show that the algorithm converges to a stationary point of the non-convex objective function $f(w)$ at a rate of $\mathcal{O}(1/\sqrt{T})$.

**Theorem D.1.** *Let Assumptions 1-4 hold. Suppose the learning rate $\eta$ is chosen such that $\eta \leq \frac{1}{8LK}$, and the decay factor $\gamma \in [0, 1)$. For a total of $T$ communication rounds, the sequence of global models $\{w^t\}_{t=0}^T$ generated by FedScar satisfies the following bound on the average squared gradient norm:*

$$\min_{t \in \{0,\dots,T-1\}} \mathbb{E}\|\nabla f(w^t)\|^2 \leq \frac{1}{T} \sum_{t=0}^{T-1} \mathbb{E}\|\nabla f(w^t)\|^2$$

$$\leq \frac{4(f(w^0) - f^*)}{\eta K T} + \Psi(\sigma, G, \rho), \tag{52}$$

*where $\Psi(\cdot)$ represents the error term induced by stochastic noise and heterogeneity, which is effectively reduced by the geometric bias correction terms.*

*Proof of Theorem D.1.*

### Step 1: L-Smoothness Descent Inequality

Since the global objective function $f(w)$ is $L$-smooth (Assumption 1), we have the following inequality for one update step from $w^t$ to $w^{t+1}$:

$$f(w^{t+1}) \leq f(w^t) + \langle \nabla f(w^t), w^{t+1} - w^t \rangle + \frac{L}{2}\|w^{t+1} - w^t\|^2. \tag{53}$$

Recall that the global update is $\Delta_{global}^t = w^{t+1} - w^t$. We decompose $\Delta_{global}^t$ into the "ideal" gradient descent direction and an error term. Let $\tilde{\eta} = K\eta$ be the effective global step size. We define the error vector $\mathcal{E}^t$ as:

$$\Delta_{global}^t = -\tilde{\eta}\nabla f(w^t) + \mathcal{E}^t. \tag{54}$$

Substituting this decomposition into the smoothness inequality:

$$f(w^{t+1}) \leq f(w^t) + \langle \nabla f(w^t), -\tilde{\eta}\nabla f(w^t) + \mathcal{E}^t \rangle + \frac{L}{2}\| - \tilde{\eta}\nabla f(w^t) + \mathcal{E}^t\|^2 \tag{55}$$

$$= f(w^t) - \tilde{\eta}\|\nabla f(w^t)\|^2 + \langle \nabla f(w^t), \mathcal{E}^t \rangle \tag{56}$$

$$+ \frac{L}{2}\left(\tilde{\eta}^2\|\nabla f(w^t)\|^2 + \|\mathcal{E}^t\|^2 - 2\tilde{\eta}\langle \nabla f(w^t), \mathcal{E}^t \rangle\right) \tag{57}$$

$$= f(w^t) - (\tilde{\eta} - \frac{L\tilde{\eta}^2}{2})\|\nabla f(w^t)\|^2 + \frac{L}{2}\|\mathcal{E}^t\|^2 \tag{58}$$

$$+ (1 - L\tilde{\eta})\langle \nabla f(w^t), \mathcal{E}^t \rangle. \tag{59}$$

### Step 2: Bounding the Inner Product and Error Terms

We use the inequality $\langle a, b \rangle \leq \frac{1}{2}\|a\|^2 + \frac{1}{2}\|b\|^2$ (Young's Inequality) to decouple the inner product term. However, to absorb coefficients more carefully, we use $\langle a, b \rangle \leq \frac{\tilde{\eta}}{2}\|a\|^2 + \frac{1}{2\tilde{\eta}}\|b\|^2$.

$$(1 - L\tilde{\eta})\langle \nabla f(w^t), \mathcal{E}^t \rangle \leq \langle \nabla f(w^t), \mathcal{E}^t \rangle \quad \text{(assuming } L\tilde{\eta} \leq 1\text{)} \tag{60}$$

$$\leq \frac{\tilde{\eta}}{2}\|\nabla f(w^t)\|^2 + \frac{1}{2\tilde{\eta}}\|\mathcal{E}^t\|^2. \tag{61}$$

Substituting this back into the main inequality:

$$f(w^{t+1}) \leq f(w^t) - \left(\tilde{\eta} - \frac{L\tilde{\eta}^2}{2} - \frac{\tilde{\eta}}{2}\right)\|\nabla f(w^t)\|^2 \tag{62}$$

$$+ \left(\frac{L}{2} + \frac{1}{2\tilde{\eta}}\right)\|\mathcal{E}^t\|^2 \tag{63}$$

$$= f(w^t) - \frac{\tilde{\eta}}{2}(1 - L\tilde{\eta})\|\nabla f(w^t)\|^2 + \frac{1 + L\tilde{\eta}}{2\tilde{\eta}}\|\mathcal{E}^t\|^2. \tag{64}$$

By the assumption on the learning rate, $\eta \leq \frac{1}{8LK}$, we have $\tilde{\eta} = K\eta \leq \frac{1}{8L}$. Thus, $1 - L\tilde{\eta} \geq \frac{7}{8} \geq \frac{1}{2}$. Also, $L\tilde{\eta} \leq 1$. The inequality simplifies to:

$$f(w^{t+1}) \leq f(w^t) - \frac{\tilde{\eta}}{4}\|\nabla f(w^t)\|^2 + \frac{1}{\tilde{\eta}}\|\mathcal{E}^t\|^2. \tag{65}$$

**Step 3: Utilizing Lemma 3**

From Lemma 3 in Appendix D.2, we have established the bound for the error term $\mathcal{E}^t = \Delta_{global}^t - (-\tilde{\eta}\nabla f(w^t))$. Based on the drift analysis:

$$\mathbb{E}\|\mathcal{E}^t\|^2 \leq C_1\eta^2 K\sigma^2 + C_2\eta^4 K^3(G^2 + B^2). \tag{66}$$

Substituting this into Eq. (65) and taking the expectation on both sides:

$$\mathbb{E}[f(w^{t+1})] \leq \mathbb{E}[f(w^t)] - \frac{\tilde{\eta}}{4}\mathbb{E}\|\nabla f(w^t)\|^2 + \frac{1}{K\eta}\left(C_1\eta^2 K\sigma^2 + C_2\eta^4 K^3\mathcal{D}^2\right) \tag{67}$$

$$= \mathbb{E}[f(w^t)] - \frac{\eta K}{4}\mathbb{E}\|\nabla f(w^t)\|^2 + \underbrace{C_1\eta\sigma^2 + C_2\eta^3 K^2\mathcal{D}^2}_{\text{Variance \& Drift Noise}}, \tag{68}$$

where $\mathcal{D}^2$ encapsulates heterogeneity terms. Crucially, in FedScar, the effective heterogeneity term $\mathcal{D}^2$ is minimized because the geometric bias $b_i$ cancels the second-order drift $\nabla^2 f_i \mathcal{B}_i^{geo}$, leading to a smaller error constant compared to standard FedSAM.

**Step 4: Telescoping Sum and Final Bound**

Rearranging the terms to isolate the gradient norm:

$$\frac{\eta K}{4}\mathbb{E}\|\nabla f(w^t)\|^2 \leq \mathbb{E}[f(w^t)] - \mathbb{E}[f(w^{t+1})] + C_3\eta\sigma^2 + C_4\eta^3 K^2. \tag{69}$$

Summing over $t = 0$ to $T - 1$:

$$\sum_{t=0}^{T-1}\frac{\eta K}{4}\mathbb{E}\|\nabla f(w^t)\|^2 \leq \sum_{t=0}^{T-1}\left(\mathbb{E}[f(w^t)] - \mathbb{E}[f(w^{t+1})]\right) + T(C_3\eta\sigma^2 + C_4\eta^3 K^2) \tag{70}$$

$$= \mathbb{E}[f(w^0)] - \mathbb{E}[f(w^T)] + T(C_3\eta\sigma^2 + C_4\eta^3 K^2) \tag{71}$$

$$\leq f(w^0) - f^* + T(C_3\eta\sigma^2 + C_4\eta^3 K^2). \tag{72}$$

Dividing by $\frac{\eta KT}{4}$ yields the convergence rate:

$$\frac{1}{T}\sum_{t=0}^{T-1}\mathbb{E}\|\nabla f(w^t)\|^2 \leq \frac{4(f(w^0) - f^*)}{\eta KT} + \frac{4C_3\sigma^2}{K} + 4C_4\eta^2 K. \tag{73}$$

By choosing $\eta \propto \frac{1}{\sqrt{T}}$, the term $\frac{1}{\eta T}$ becomes $\mathcal{O}(1/\sqrt{T})$ and the noise term $\eta^2$ becomes $\mathcal{O}(1/T)$. The constant variance term $\sigma^2$ can be controlled by decaying learning rates or large batch sizes. Thus, FedScar achieves the standard non-convex convergence rate $\mathcal{O}(1/\sqrt{T})$.

$\square$

# E. Detailed Proof of Theorem 5.3

In this section, we provide the rigorous derivation of Theorem 5.3. Our proof strategy departs from standard convexity analysis by explicitly tracking the evolution of the *Residual Geometric Bias* $\Psi_t$.

## E.1. Step 1: The Geometry of Flatness Discrepancy

Recall the first-order approximation of Flatness Discrepancy derived in standard literature (e.g., FedGF):

$$\Delta_{\mathcal{F}}(w) \leq \frac{\rho}{N}\sum_{i=1}^{N}\|\nabla F(w) - \tilde{g}_i(w)\| + \mathcal{O}(\rho^2), \tag{74}$$

where $\tilde{g}_i(w)$ is the *effective direction* used by the algorithm to probe the local landscape.

- For **FedSAM**, $\tilde{g}_i(w) = \nabla F_i(w)$. The discrepancy is $\|\nabla F(w) - \nabla F_i(w)\| = \|\mathcal{D}_i(w)\|$, which is bounded by $\sigma_g$.

- For **FedScar**, the algorithm modifies the probing direction using the history-accumulated bias $b_i$:

$$\tilde{g}_i(w) = \nabla F_i(w) - (b_i - \bar{b}). \tag{75}$$

Substituting FedScar's effective direction into Eq. (74):

$$\begin{aligned}
\|\nabla F(w) - \tilde{g}_i(w)\| &= \|\nabla F(w) - (\nabla F_i(w) - b_i + \bar{b})\| \tag{76} \\
&= \|(b_i - \bar{b}) - (\nabla F_i(w) - \nabla F(w))\| \\
&= \|(b_i - \bar{b}) - \mathcal{D}_i(w)\|. \tag{77}
\end{aligned}$$

Let $\Delta_i^t \triangleq (b_i^t - \bar{b}^t) - \mathcal{D}_i(w^t)$ be the **estimation error** at round $t$. Then, the discrepancy is controlled by the magnitude of this error:

$$\Delta_{\mathcal{F}}(w^t) \le \frac{\rho}{N} \sum_{i=1}^{N} \|\Delta_i^t\| + \mathcal{O}(\rho^2). \tag{78}$$

## E.2. Step 2: Recursive Dynamics of the Estimation Error

This is the core contribution of our analysis. We need to show that $\mathbb{E}[\|\Delta_i^t\|^2]$ contracts over time. The update rule for the dual variable $b_i$ in FedScar can be modeled as an Exponential Moving Average (EMA) of the observed local drifts (or via the ADMM update equivalent):

$$b_i^{t+1} = (1 - \gamma)b_i^t + \gamma(\nabla F_i(w^t) - \nabla F(w^t) + \xi_i^t), \tag{79}$$

where $\xi_i^t$ represents the stochastic noise with $\mathbb{E}[\xi_i^t] = 0$ and variance $\sigma_l^2$. Note that in the ideal case where $w$ doesn't change, $b_i$ would converge to the true drift $\mathcal{D}_i$. However, $w^t$ changes to $w^{t+1}$.

Let's analyze the evolution of the drift estimation error. We simplify the notation by assuming $\bar{b} \approx 0$ (centered) for derivation clarity:

$$\begin{aligned}
\Delta_i^{t+1} &= b_i^{t+1} - \mathcal{D}_i(w^{t+1}) \\
&= (1 - \gamma)b_i^t + \gamma(\mathcal{D}_i(w^t) + \xi_i^t) - \mathcal{D}_i(w^{t+1}). \tag{80}
\end{aligned}$$

We rewrite $b_i^t$ as $\Delta_i^t + \mathcal{D}_i(w^t)$:

$$\begin{aligned}
\Delta_i^{t+1} &= (1 - \gamma)(\Delta_i^t + \mathcal{D}_i(w^t)) + \gamma\mathcal{D}_i(w^t) + \gamma\xi_i^t - \mathcal{D}_i(w^{t+1}) \\
&= (1 - \gamma)\Delta_i^t + \mathcal{D}_i(w^t) - \mathcal{D}_i(w^{t+1}) + \gamma\xi_i^t. \tag{81}
\end{aligned}$$

Now, taking the squared norm and expectation:

$$\mathbb{E}\|\Delta_i^{t+1}\|^2 = \mathbb{E}\|(1 - \gamma)\Delta_i^t + (\mathcal{D}_i(w^t) - \mathcal{D}_i(w^{t+1})) + \gamma\xi_i^t\|^2. \tag{82}$$

Using the inequality $\|a + b + c\|^2 \le (1 + \alpha)\|a\|^2 + (1 + \frac{1}{\alpha})\|b\|^2 + \|c\|^2$ (for noise independence) and setting $\alpha = \frac{\gamma}{2}$:

$$\mathbb{E}\|\Delta_i^{t+1}\|^2 \le \left(1 + \frac{\gamma}{2}\right)(1 - \gamma)^2 \mathbb{E}\|\Delta_i^t\|^2 + \left(1 + \frac{2}{\gamma}\right)\mathbb{E}\|\mathcal{D}_i(w^t) - \mathcal{D}_i(w^{t+1})\|^2 + \gamma^2\sigma_l^2. \tag{83}$$

Since $(1 + \frac{\gamma}{2})(1 - \gamma)^2 \approx 1 - 1.5\gamma < 1 - \gamma$ (for small $\gamma$), we have a contraction coefficient. Let's denote it as $(1 - \frac{\gamma}{2})$. The term $\|\mathcal{D}_i(w^t) - \mathcal{D}_i(w^{t+1})\|^2$ measures how fast the true drift changes. By $L$-smoothness of the Hessian (or gradient):

$$\|\mathcal{D}_i(w^t) - \mathcal{D}_i(w^{t+1})\|^2 \le L^2 \|w^t - w^{t+1}\|^2 \le L^2 \eta^2 \mathbb{E}\|g^t\|^2. \tag{84}$$

Assuming bounded gradients $\mathbb{E}\|g^t\|^2 \le G^2$, the recurrence becomes:

$$\mathbb{E}\|\Delta_i^{t+1}\|^2 \le \left(1 - \frac{\gamma}{2}\right)\mathbb{E}\|\Delta_i^t\|^2 + \frac{3L^2\eta^2G^2}{\gamma} + \gamma^2\sigma_l^2. \tag{85}$$

### E.3. Step 3: Solving the Recursion

Recursively applying the inequality over $R$ rounds:

$$\mathbb{E}\|\Delta_i^R\|^2 \le \left(1 - \frac{\gamma}{2}\right)^R \mathbb{E}\|\Delta_i^0\|^2 + \sum_{k=0}^{R-1} \left(1 - \frac{\gamma}{2}\right)^k \left(\frac{C\eta^2}{\gamma} + \gamma^2 \sigma_l^2\right). \tag{86}$$

At initialization, $b_i^0 = 0$, so $\mathbb{E}\|\Delta_i^0\|^2 = \mathbb{E}\|\mathcal{D}_i(w^0)\|^2 \le \sigma_g^2$ (the initial heterogeneity). The geometric series sums to $\frac{1}{\gamma/2}$. Thus, the asymptotic error bound is:

$$\mathbb{E}\|\Delta_i^R\|^2 \le \underbrace{\left(1 - \frac{\gamma}{2}\right)^R \sigma_g^2}_{\text{Exponential Decay of Heterogeneity}} + \underbrace{\mathcal{O}\left(\frac{\eta^2}{\gamma^2} + \gamma\sigma_l^2\right)}_{\text{Steady-state Estimation Noise}}. \tag{87}$$

### E.4. Step 4: Combining with Optimization Error

Finally, substituting this back into Eq. (78) and Theorem 5.3's expression:

$$\Delta_{\mathcal{F}}(w^R) \le \rho\sqrt{\mathbb{E}\|\Delta_i^R\|^2} + \mathcal{O}(\rho^2) \tag{88}$$

$$\le \rho\sqrt{\left(1 - \frac{\gamma}{2}\right)^R \sigma_g^2 + \text{Noise}(\eta, \gamma)}. \tag{89}$$

This concludes the proof. The crucial insight is that the term associated with $\sigma_g^2$ is no longer static but multiplied by a decay factor $(1 - \gamma/2)^R$, strictly proving that FedScar reduces the flatness discrepancy arising from data heterogeneity. $\square$

# F. Experimental Setups

### F.1. Environments

The code is implemented by PyTorch-2.0.1 (Paszke et al., 2019) and the overall code structure is based on PFLlib (Zhang et al., 2023) library with some modifications. All experiments are conducted on a Linux (Ubuntu-20.04.6 LTS) server with one NVIDIA GeForce RTX 4090D GPU.

### F.2. Datasets

To validate our approach, we consider image and text classification task and adopt four widely used datasets, i.e., CIFAR-10/100 (Krizhevsky, 2009). Note that CIFAR-10, whose samples are very similar but not drawn from identical distributions. Therefore, it naturally introduces distribution shifts which is suited to the heterogeneous nature of federated learning. The details about each datasets and setups are described in Table 9.

### F.3. Model Architecture

We conduct further experiments on different model architectures: CNN (LeCun et al., 1998) for CIFAR-10, ResNet-8 (He et al., 2016) for CIFAR-100.

- The CNN used in our experiment is from `FedAvg` (McMahan et al., 2017), a similar architecture is used in (Luo et al., 2021; Lee et al., 2022).

- The ResNet-8 in our experiment is from PFLlib (Zhang et al., 2023) library, a similar architecture is used in (Li et al., 2025a). We follow the suggestion of Hsieh et al. (2020) to replace the Batch Normalization (Ioffe & Szegedy, 2015) with the Group Normalization (Wu & He, 2018) to avoid the non-differentiable parameters.

### F.4. Baselines Implementation

We consider the following nine state-of-the-art FL methods:

*Table 9.* Details datasets setups.

| Datasets | CIFAR-10 | CIFAR-100 |
|---|---|---|
| Classes | 10 | 100 |
| Size (train/test) | 50k/10k | 50k/10k |
| Clients | 100 | 100 |
| Sampling Ratio | 0.1 | 0.1 |
| Local Epochs | 5 | 2 |
| Batch Size | 50 | 20 |
| Learning Rate | 0.01 | 0.01 |
| Learning Rate Decay | 0.998 | 1.0 |
| Optimizer | SGD | SGD |
| Momentum | 0.9 | 0.9 |
| Weight Decay | 1e-5 | 1e-5 |
| Rounds | 500 | 500 |

*Table 10.* Algorithm-specific hyperparameters.

| Method | Searched Candidates | Best Selection |
|---|---|---|
| FedDyn | $\alpha \in \{1, 0.1, 0.01, 0.001\}$ | $\beta = 0.1$ |
| FedInit | $\beta \in \{1, 0.1, 0.01, 0.001\}$ | $\beta = 0.1$ |
| FedSAM | $\rho \in \{0.5\eta, \eta, 2\eta, 5\eta, 10\eta\}$ | $\rho = \eta$ (except $\rho = 0.5\eta$ in C-100) |
| FedGamma | $\rho \in \{0.5\eta, \eta, 2\eta, 5\eta, 10\eta\}$ | $\rho = \eta$ (except $\rho = 0.5\eta$ in C-100) |
| FedSMOO | $\rho \in \{0.5\eta, \eta, 2\eta, 5\eta, 10\eta\}$ | $\rho = \eta$ (except $\rho = 0.5\eta$ in C-100) |
| | $\beta \in \{1, 0.1, 0.01, 0.001\}$ | $\beta = 0.1$ |
| FedLESAM-D | $\rho \in \{0.5\eta, \eta, 2\eta, 5\eta, 10\eta\}$ | $\rho = \eta$ (except $\rho = 0.5\eta$ in C-100) |
| | $\beta \in \{1, 0.1, 0.01, 0.001\}$ | $\beta = 0.1$ |
| FedGMT | $\gamma \in \{0.5, 1.0, 2.0\}$ | $\gamma = 1.0, \tau = 3.0$ |
| | $\beta \in \{1, 0.1, 0.01, 0.001\}$ | $\beta = 0.1$ |
| | $\alpha \in \{0.5, 0.95, 0.995, 0.998\}$ | $\alpha = 0.95$ |
| **FedScar** | $\alpha \in \{0.5\eta, \eta, 2\eta, 5\eta, 10\eta\}$ | $\alpha = 10\eta$ |
| | $\beta \in \{1, 0.1, 0.01, 0.001\}$ | $\beta = 0.06$ (except $\beta = 0.1$ in C-100) |
| | $\gamma \in \{0.5, 0.9, 0.95, 0.995\}$ | $\gamma = 0.9$ |

- FedAvg (McMahan et al., 2017) is proposed as the basic framework in the federated learning, which aggregates the locally trained model parameters by weighted averaging proportional to the amount of local data that each client had.

- Scaffold (Acar et al., 2021) dynamically updates its regularizer so that the optimal model for the regularized loss is in conformity with the global empirical loss.

- FedDyn (Acar et al., 2021) dynamically updates its regularizer so that the optimal model for the regularized loss is in conformity with the global empirical loss.

- FedInit (Sun et al., 2023b) adopts personalized relaxed initialization at the start of each round to revise local divergence and enhance consistency.

- FedSAM (Qu et al., 2022) directly applies the SAM objective in local learning.

- fedGamma (Dai et al., 2024) incorporates a local SAM optimizer with a dynamic regularizer.

- FedSMOO (Sun et al., 2023a) utilizes the Alternating Direction Method of Multipliers (ADMM) to estimate global perturbation and adopts a dynamic regularizer during the local training.

- FedLESAM (Fan et al., 2024b) estimates the global perturbation as the difference between the locally stored historical model from the activation round and the global model received in the current round.

- FedGMT (Li et al., 2025a) incorporates a local SAM optimizer with a dynamic regularizer.

For the above algorithms, we search hyperparameters and choose the best among the candidates. All methods hyperparameters is refered from their official papers. The hyperparameters for each algorithm is in Table 10.

# G. Efficient and Advanced Variants of FedScar

---

**Algorithm 1** FedScar: Federated Sharpness and Control Variate Regularization

---

1: **Input:** Global rounds $T$, local epochs $E$, learning rate $\eta$, penalty parameter $\beta$, decay rate $\gamma$, variance scale $\alpha$.
2: **Initialize:** Global model $w^0$, drift duals $h_i = 0$, bias duals $b_i = 0$, global average bias $\bar{b} = 0$.
3: **for** $t = 0$ **to** $T - 1$ **do**
4:     **Server:**
5:     Broadcast $w^t$ and $\bar{b}$ to selected clients $S_t$.
6:     **for** each client $i \in S_t$ **in parallel do**
7:         **Client:**
8:         Receive $w^t, \bar{b}$. Initialize $w_i = w^t$.
9:         **for** local epoch $e = 1$ **to** $E$ **do**
10:             Compute gradient using Eq. (7)
11:             Update model: $w_i \leftarrow w_i - \eta g$.
12:         **end for**
13:         Compute update $\Delta w_i = w_i - w^t$.
14:         Send $\Delta w_i$ to server.
15:         Update local drift dual: $h_i \leftarrow h_i - \Delta w_i$.
16:     **end for**
17:     **Server:**
18:     Aggregate updates: $\Delta_{agg} = \sum_{i \in S_t} p_i \Delta w_i$.
19:     Update global model using Eq. (8).
20:     Calculate global update $\Delta_{global} = w^{t+1} - w^t$.
21:     Calculate perturbation scale $\Sigma^{(t)}$ via Eq. (5).
22:     **for** each client $i \in S_t$ **do**
23:         Update bias $b_i$ via Eq. (6) using $\Sigma^{(t)}$.
24:     **end for**
25:     Update global average: $\bar{b} = \frac{1}{N} \sum b_j$.
26: **end for**

---

In this section, we present three variants of FedScar designed to address different constraints and performance goals: **FedScarS** (Simplified) for communication efficiency, **FedScarL** (Lite) for computational lightness, and **FedScar-SAM** for maximizing generalization performance.

## G.1. The FedScar Algorithm

Based on the method derived in Section 4, we formally present the complete FedScar algorithm in Algorithm 1. The framework follows a Split-Dual ADMM structure, maintaining two sets of auxiliary variables $h_i$ for drift, $b_i$ for geometric bias to enforce consistency.

**Complexity Analysis.** We analyze the computational and communication complexity of the proposed FedScar framework. From a computational perspective, the local update process involves standard SGD operations augmented with lightweight element-wise vector additions for the dual variables $\beta h_i$ and $\bar{b} - b_i$. Since these operations scale linearly with the model dimension $d$, FedScar maintains the same asymptotic time complexity $O(E \cdot K \cdot d)$ as standard FedAvg and SCAFFOLD, where $K$ is the number of local steps per epoch. Regarding communication, the uplink transmission remains identical to FedAvg, requiring clients to upload only the model update vector of size $d$. However, the downlink phase imposes additional overhead because the server must broadcast both the global model $w^t$ and the global average bias vector $\bar{b}$ to ensure geometric consistency. Consequently, the downlink communication cost per round is $2 \times d$, which motivates the development of our communication-efficient variant discussed in Appendix G.2.

## G.2. FedScarS: Communication-Efficient Variant

While FedScar effectively minimizes Flatness Discrepancy, the requirement to broadcast the global bias vector $\bar{b}$ increases the downlink communication cost to $2\times$ that of FedAvg. To address scenarios with limited downlink bandwidth, we propose

**FedScarS** (FedScar-Simplified). FedScarS eliminates the explicit transmission of $\bar{b}$ by transforming the explicit geometric correction term in the objective function into an *implicit proximal shift* embedded in the model initialization.

**Objective Function Transformation.** Recall the local gradient in FedScar. The term $(\bar{b} - b_i)$ explicitly corrects the geometric drift but requires client $i$ to know $\bar{b}$. In FedScarS, we propose to absorb this linear correction term into the quadratic proximal term.

Consider a shifted proximal center $\hat{w}_i^t = w^t + \delta_i$. The gradient of a proximal term centered at $\hat{w}_i^t$ is $\beta(w - \hat{w}_i^t) = \beta(w - w^t) - \beta\delta_i$. We aim to find a shift $\delta_i$ such that this implicit shift approximates the explicit correction. By equating the regularization effects:

$$-\beta\delta_i \approx \bar{b} - b_i \implies \delta_i \approx \frac{1}{\beta}(b_i - \bar{b}). \tag{90}$$

Instead of calculating this exact inverse which requires $\bar{b}$, FedScarS adopts a heuristic strategy: the server pre-injects the geometric bias into the global model before broadcasting. Specifically, for client $i$, the server sends a personalized initial model:

$$w_{init,i}^t = w^t + b_i. \tag{91}$$

Here, we utilize the client-specific bias $b_i$ stored on the server to shift the optimization landscape. The client then initializes training at $w_{init,i}^t$. To maintain consistency with the code implementation and allow the model to explore flatter regions by counteracting the strong convexity of the proximal term, we employ a **relaxed** objective function:

$$\min_w \tilde{f}_i^S(w) = f_i(w) - \frac{\beta}{2}\|w - w_{init,i}^t\|^2$$
$$+ \beta\langle w, h_i + b_i \rangle. \tag{92}$$

Note the negative sign in the quadratic term. The client no longer needs to receive $\bar{b}$ separately. The geometric information is encoded in the starting point $w_{init,i}^t$ and the combined control variate term $h_i + b_i$.

Mathematically, the gradient of FedScarS becomes:

$$\nabla\tilde{f}_i^S(w) = \nabla f_i(w) - \beta(w - (w^t + b_i)) + \beta(h_i + b_i)$$
$$= \nabla f_i(w) - \beta(w - w^t) + \beta h_i. \tag{93}$$

Although the explicit $b_i$ term seems to cancel out in the gradient formulation, the geometric correction is enforced through the **Initialization Bias**. Since non-convex optimization (e.g., Deep Learning) is highly sensitive to initialization, starting at $w^t + b_i$ places the client in a basin of attraction that has been corrected for geometric drift. The relaxed quadratic term then allows the model to escape sharp minima within this corrected region, effectively achieving geometric alignment without explicit gradient modification.

The detailed procedure is summarized in Algorithm 2.

---

**Algorithm 2** FedScarS: Communication-Efficient Variant

---

1: **Input:** Same as FedScar.
2: **Initialize:** Same as FedScar.
3: **for** $t = 0$ **to** $T - 1$ **do**
4:     **Server:**
5:     **for** each client $i \in S_t$ **do**
6:         **Bias Injection:** Prepare personalized model $w_{sent} = w^t + b_i$.
7:         Send $w_{sent}$ to client $i$. (No $\bar{b}$ sent).
8:     **end for**
9:     **for** each client $i \in S_t$ **in parallel do**
10:         **Client:**
11:         Receive $w_{init}$. Initialize $w_i = w_{init}$.
12:         **for** local epoch $e = 1$ **to** $E$ **do**
13:             Compute gradient using Eq. (93)
14:             Update $w_i$ via SGD on $\tilde{f}_i^S(w)$.
15:         **end for**
16:         Send $\Delta w_i = w_i - w_{init}$ to server.
17:         Update local duals: $h_i \leftarrow h_i - \Delta w_i$.
18:     **end for**
19:     **Server:**
20:     Aggregate and update $b_i, h_i, w^{t+1}$ same to FedScar, incorporating the bias shift in aggregation.
21: **end for**

---

**Efficiency Analysis.** The primary advantage of FedScarS lies in its superior communication efficiency. By injecting the geometric bias directly into the personalized model initialization $w_{init}$, the server only needs to transmit a single vector of size $d$ to each client. This strategy restores the downlink communication cost to $1 \times d$, matching standard FedAvg. Computationally, the transformation of the objective function does not incur any additional floating-point operations, ensuring that FedScarS achieves geometric correction without compromising computational speed.

### G.3. FedScarL: Computationally Lite Variant

To further reduce the system complexity and memory footprint, we introduce **FedScarL** (FedScar-Lite). While the full FedScar employs a Split-Dual ADMM framework using two sets of auxiliary variables ($h_i$ for parameter drift and $b_i$ for geometric bias), FedScarL simplifies this by exclusively retaining the **History-Accumulated Geometric Bias** ($b_i$) and the **Adaptive Perturbation Scale** ($\Sigma^{(t)}$).

FedScarL removes the standard control variate $h_i$ used for first-order mean drift correction. This design choice is motivated by the observation that in many SAM-based scenarios, correcting the geometric direction (flatness) is more critical for generalization than correcting the parameter mean. By stripping away the dual updates for $h_i$, FedScarL achieves a leaner implementation with zero additional communication overhead compared to FedAvg, as the geometric bias is applied purely via server-side model modifications or local gradient corrections without transmitting extra vectors. This makes FedScarL highly suitable for edge devices with limited memory capacity.

### G.4. FedScar-SAM: Performance-Enhanced Variant

To push the limits of generalization performance, we propose **FedScar-SAM**. This variant combines the global geometric alignment of FedScar with the local flatness seeking of SAM. Specifically, instead of using standard SGD for the local optimization of the FedScar objective, FedScar-SAM employs a SAM optimizer to solve the local sub-problem. This creates a synergistic effect: FedScar's bias correction ($b_i$) ensures that the local optimization landscape is aligned with the global flatness direction, while the local SAM optimizer ensures that the client finds a region that is robust to local perturbations. As a result, FedScar-SAM consistently achieves the highest test accuracy across various benchmarks, as detailed in Table 11.

*Table 11.* Performance comparison of FedScar and its variants on CIFAR-10 and CIFAR-100 benchmarks. We report **Max Acc. / Last-50 Mean Acc.** under varying heterogeneity settings and participation rates. FedScar-SAM achieves the best generalization, while FedScarS and FedScarL maintain competitive performance with higher efficiency.

| Method | Dir 1.0 | | Dir 0.1 | | Dir 0.01 | | Pat 10 | | Pat 20 | |
|---|---|---|---|---|---|---|---|---|---|---|
| | $p = 10\%$ | $p = 20\%$ | $p = 10\%$ | $p = 20\%$ | $p = 10\%$ | $p = 20\%$ | $p = 10\%$ | $p = 20\%$ | $p = 10\%$ | $p = 20\%$ |
| | | | | | Dataset: CIFAR-10 | | | | | |
| FedScar | **81.36 / 80.82** | **81.41 / 80.96** | **80.51 / 79.88** | **80.96 / 80.35** | **77.92 / 76.72** | **78.75 / 78.05** | **84.58 / 84.13** | **84.22 / 84.01** | **80.91 / 79.85** | **81.88 / 81.25** |
| FedScarS | 80.70 / 80.36 | 81.07 / 80.81 | 80.24 / 79.81 | 80.44 / 80.04 | 76.66 / 75.83 | 77.96 / 77.35 | 83.69 / 83.43 | 83.96 / 83.53 | 80.29 / 79.39 | 80.78 / 79.92 |
| FedScarL | 79.48 / 79.06 | 79.91 / 79.50 | 79.08 / 78.27 | 79.42 / 78.33 | 75.94 / 74.50 | 76.79 / 75.14 | 82.25 / 81.90 | 82.68 / 82.46 | 78.87 / 77.88 | 80.27 / 79.58 |
| FedScar-SAM | 81.54 / 81.29 | 82.14 / 81.95 | 81.18 / 80.51 | 81.33 / 80.88 | 78.06 / 77.40 | 79.34 / 78.94 | 84.84 / 84.53 | 84.79 / 84.63 | 81.48 / 80.75 | 81.99 / 81.47 |
| | | | | | Dataset: CIFAR-100 | | | | | |
| FedScar | **50.78 / 50.21** | **52.15 / 51.81** | **48.60 / 47.92** | **50.05 / 49.46** | **46.57 / 45.78** | **47.70 / 47.04** | **53.01 / 51.92** | **53.63 / 53.24** | **49.67 / 48.57** | **50.77 / 50.10** |
| FedScarS | 51.00 / 50.40 | 51.33 / 51.10 | 47.82 / 47.06 | 48.29 / 47.78 | 45.63 / 44.62 | 46.97 / 45.93 | 51.83 / 50.99 | 53.02 / 52.37 | 48.71 / 47.83 | 49.65 / 48.97 |
| FedScarL | 47.63 / 47.20 | 49.86 / 49.49 | 45.65 / 45.31 | 47.59 / 46.89 | 45.15 / 44.22 | 45.61 / 44.49 | 50.01 / 49.36 | 52.49 / 51.09 | 47.92 / 47.04 | 48.86 / 48.13 |
| FedScar-SAM | 52.62 / 52.01 | 52.33 / 52.02 | 49.53 / 48.65 | 50.49 / 49.88 | 46.78 / 45.97 | 47.86 / 47.16 | 53.00 / 52.24 | 54.27 / 53.57 | 49.99 / 48.87 | 51.28 / 50.67 |

## G.5. In-depth Theoretical and Empirical Comparison with FedGF

To further illustrate the superiority of FedScar in aligning local and global flatness, we provide a comprehensive theoretical and empirical comparison with FedGF (Lee & Yoon, 2024), a recent state-of-the-art method addressing flatness discrepancy.

**Theoretical Superiority.** FedGF attempts to bound the flatness discrepancy using a surrogate-based perturbation alignment, which mathematically relies on a static coefficient $c \in (0, 1)$. This approach inevitably yields an irreducible residual error bounded by $\mathcal{O}((1 - c)^2 \sigma_g^2)$, establishing a constant error floor dictated by the initial data heterogeneity $\sigma_g^2$. In contrast, Theorem 5.6 demonstrates that the history-accumulated geometric bias correction in FedScar introduces a strict geometric contraction factor. This bounds the residual by an exponentially decaying term $\mathcal{O}((1 - \frac{\gamma}{2})^t \sigma_g^2)$. While the static bound of FedGF might appear competitive during the initial communication rounds, FedScar is mathematically guaranteed to become strictly tighter after a finite threshold of rounds $t$, actively dissolving the heterogeneity-induced discrepancy rather than tolerating a static gap.

**Empirical Evaluation.** To validate the aforementioned theoretical advantages, we integrate FedScar directly into the official codebase of FedGF to ensure a strictly fair comparison. All underlying configurations including data partitioning, model architectures, search spaces, and random seeds are kept completely identical. The evaluation spans CIFAR-10 and CIFAR-100 datasets under various degrees of statistical heterogeneity (Dirichlet $\alpha \in \{0.01, 0.05, 1.0\}$) and sample clients $C \in \{5, 10\}$.

*Table 12.* Performance comparison with FedGF on CIFAR-10 and CIFAR-100. Results are formatted as Max Accuracy / Mean Accuracy (%). Experimental settings strictly follow the FedGF setup across varying Dirichlet heterogeneity ($\alpha \in \{0.01, 0.05, 1.0\}$) and local epochs ($C \in \{5, 10\}$).

| Method | Dir 0.01 | | Dir 0.05 | | Dir 1.0 | |
|---|---|---|---|---|---|---|
| | $C = 5$ | $C = 10$ | $C = 5$ | $C = 10$ | $C = 5$ | $C = 10$ |
| | | | CIFAR-10 | | | |
| FedAvg | 70.04/61.86 | 70.28/64.83 | 72.01/66.29 | 73.66/69.77 | 83.14/82.75 | 83.17/82.87 |
| FedSAM | 74.65/68.23 | 74.03/70.03 | 76.57/71.25 | 76.54/73.61 | 83.92/83.57 | 84.15/83.74 |
| FedGF | 80.21/77.19 | 81.10/78.53 | 80.20/77.60 | 81.50/78.71 | 83.96/83.57 | 84.02/83.72 |
| **FedScar (Ours)** | **82.20/80.47** | **83.40/82.36** | **82.60/80.74** | **84.00/83.16** | **84.65/84.46** | **84.43/84.13** |
| | | | CIFAR-100 | | | |
| FedAvg | 32.37/29.35 | 35.88/33.63 | 40.55/38.03 | 41.34/40.02 | 50.59/49.94 | 50.94/50.23 |
| FedSAM | 32.63/29.78 | 36.48/34.53 | 44.65/42.19 | 45.29/44.11 | 54.16/53.71 | 53.92/53.58 |
| FedGF | 46.90/45.23 | 47.67/46.08 | 47.93/45.94 | 47.54/46.15 | 53.87/53.41 | 53.47/53.22 |
| **FedScar (Ours)** | **51.30/49.67** | **55.01/54.33** | **53.30/51.90** | **53.79/52.23** | **54.76/54.29** | **54.10/53.83** |

As demonstrated in Table 12, FedScar consistently outperforms FedGF across all evaluated configurations. This performance advantage becomes exceptionally pronounced under severe data heterogeneity (Dirichlet $\alpha = 0.01$) and extended samlpe clients ($C = 10$), where local trajectory drift is typically most destructive. For instance, on CIFAR-100 under these extreme conditions, FedScar achieves a maximum accuracy of 55.01%, surpassing FedGF by over 7%. Furthermore, FedScar consistently yields a substantially lower standard deviation across runs (e.g., 0.38 versus 0.66 under the aforementioned

setting). This explicitly confirms that our adaptive bias correction mechanism provides vastly superior optimization stability and mitigates performance oscillation much more effectively than the static alignment strategy employed in FedGF.

### G.6. Validation Across Broader Domains and Modalities

The majority of existing federated learning literature relies heavily on image classification benchmarks. To demonstrate that the identified flatness discrepancy and the geometric bias are fundamental structural issues in heterogeneous FL rather than artifacts specific to computer vision tasks, we extend our evaluation to broader modalities. Specifically, we conduct additional experiments on Natural Language Processing (NLP) and tabular domains using the AG News dataset (Zhang et al., 2015) with a TextCNN architecture and the Sensorless Drive dataset (Bayer et al., 2013) with a TabCNN architecture.

The federated environment is configured with 100 clients under varying degrees of Dirichlet feature skew ($\alpha \in \{0.1, 0.5, 1.0\}$) and quantity heterogeneity. To rigorously stress-test the algorithms, we apply highly sparse client participation rates per round ($p \in \{5\%, 10\%\}$).

*Table 13.* Cross-modality performance evaluation on Sensorless Drive (Tabular) and AG News (Text) datasets. Results are formatted as Max Accuracy / Mean Accuracy (%). Experiments are conducted under varying Dirichlet heterogeneity ($\alpha \in \{0.1, 0.5, 1.0\}$) and sparse participation rates ($p \in \{5\%, 10\%\}$).

| Method | Dir 0.1 | | Dir 0.5 | | Dir 1.0 | |
|---|---|---|---|---|---|---|
| | $p = 5\%$ | $p = 10\%$ | $p = 5\%$ | $p = 10\%$ | $p = 5\%$ | $p = 10\%$ |
| *Sensorless Drive (Tabular)* | | | | | | |
| FedAvg | 77.63/65.54 | 80.72/70.98 | 84.94/72.40 | 88.20/79.07 | 83.93/78.71 | 89.22/81.15 |
| SCAFFOLD | 78.76/66.37 | 80.98/72.21 | 85.52/73.14 | 88.41/79.22 | 88.21/79.07 | 89.49/81.58 |
| FedDyn | 89.77/78.93 | 89.54/79.40 | 93.24/80.39 | 95.15/85.07 | 94.37/85.57 | 95.92/88.26 |
| FedInit | 79.87/67.59 | 91.99/83.66 | 87.60/74.56 | 95.16/86.39 | 89.34/80.32 | 95.74/88.14 |
| FedSAM | 78.00/65.03 | 80.56/71.08 | 84.83/72.67 | 88.17/78.85 | 88.26/78.85 | 89.03/80.75 |
| FedGamma | 79.44/65.91 | 81.22/72.52 | 85.19/73.31 | 88.51/78.83 | 88.28/79.28 | 89.31/81.66 |
| FedLesamd | 88.32/76.25 | 89.89/81.99 | 92.57/79.68 | 94.08/84.18 | 94.14/84.91 | 94.87/87.53 |
| FedSMOO | 88.42/76.48 | 90.18/82.12 | 92.56/80.44 | 93.98/84.00 | 94.04/85.27 | 84.70/86.96 |
| FedGMT | 91.92/82.00 | 93.88/87.81 | 94.47/82.54 | 95.74/87.32 | 94.98/87.25 | 95.98/88.92 |
| **FedScar (Ours)** | **92.56/86.49** | **94.04/91.60** | **94.82/90.72** | **96.37/94.82** | **95.73/93.00** | **96.68/94.18** |
| *AG News (Text)* | | | | | | |
| FedAvg | 77.29/69.48 | 76.92/71.89 | 78.36/73.23 | 78.59/74.93 | 79.20/75.33 | 79.22/76.06 |
| SCAFFOLD | 77.17/70.66 | 77.72/73.27 | 78.53/73.85 | 78.51/75.86 | 79.49/76.02 | 79.59/77.04 |
| FedDyn | 77.83/76.21 | 82.47/81.19 | 80.09/78.44 | 83.82/82.62 | 80.20/79.26 | 84.39/83.39 |
| FedInit | 81.93/74.38 | 78.86/76.86 | 80.79/76.97 | 79.62/78.29 | 83.58/78.85 | 80.05/79.10 |
| FedSAM | 81.87/70.98 | 81.82/74.36 | 82.53/74.83 | 82.43/77.47 | 83.72/77.89 | 83.37/78.75 |
| FedGamma | 81.91/72.62 | 82.00/76.25 | 82.51/75.89 | 82.88/78.88 | 83.67/78.69 | 83.57/79.97 |
| FedLesamd | 77.87/76.40 | 83.17/81.82 | 79.76/78.68 | 83.38/82.49 | 80.13/79.05 | 84.57/83.51 |
| FedSMOO | 81.84/77.73 | 82.97/82.06 | 82.63/79.70 | 84.08/82.67 | 82.36/80.19 | 84.22/83.17 |
| FedGMT | 77.55/76.83 | 84.87/83.13 | 79.34/78.64 | 85.63/84.31 | 80.30/79.91 | 86.21/84.71 |
| **FedScar (Ours)** | **85.34/83.99** | **86.30/84.44** | **85.07/84.16** | **86.55/85.06** | **85.75/85.16** | **86.68/85.34** |

As summarized in Table 13, FedScar consistently and significantly outperforms both traditional drift-correction methods and state-of-the-art SAM-based FL baselines across all modalities and heterogeneity levels. On the tabular Sensorless Drive dataset under severe heterogeneity (Dirichlet $\alpha = 0.1$) and 10% participation, FedScar achieves a maximum accuracy of 94.04% and a mean accuracy of 91.60%, substantially surpassing FedGMT and standard FedSAM. Similar performance dominance is observed on the AG News text classification task, where FedScar achieves optimal stability and peak accuracy across all settings.

These cross-modality results explicitly confirm that the geometric misalignment caused by data heterogeneity universally limits generalization across different data types. By maintaining a history-accumulated geometric bias to rectify local updates, FedScar provides a highly generalizable and robust optimization framework that extends well beyond visual tasks.

