# OpenReview forum: "FedScar: Correcting Geometric Bias for Flatness-Consistent Federated Learning"
_ICML.cc/2026/Conference — ICML 2026 regular_

### Official Review · Reviewer_9RK9 · 2026-02-26

**Soundness:** 4
**Presentation:** 3
**Significance:** 4
**Originality:** 3
**Overall Recommendation:** 5
**Confidence:** 4

**Summary:**

This paper presents FedScar, a federated learning framework designed to address generalization collapse under severe statistical heterogeneity. The proposed method explicitly aligns the flatness discrepancy between clients and the server by leveraging a history-accumulated bias correction mechanism and a heterogeneity-sensitive perturbation scale. Extensive experiments on image classification benchmarks demonstrate that FedScar achieves consistent performance gains over standard federated learning and sharpness-aware minimization baselines under extreme non-IID settings, without incurring computational cost or uplink communication overhead.

**Compliance With Llm Reviewing Policy:**

Affirmed.

**Final Justification:**

This paper proposes FedScar, a federated learning framework that addresses generalization collapse under severe heterogeneity by aligning flatness discrepancies through bias correction and adaptive perturbation. Experimental results demonstrate that FedScar consistently outperforms SAM-based baselines in extreme non-IID scenarios without increasing computational or communication overhead. Since the authors have successfully addressed all my concerns during the rebuttal phase, I recommend the paper for Acceptance.

**Key Questions For Authors:**

1. Could the authors clarify the specific GPU hardware used to report the timing metrics and specify the precision format assumed for the 1.5x communication overhead reported in Table 1?
2. Could the authors briefly discuss whether the proposed framework is designed to naturally integrate with existing federated learning techniques such as gradient quantization or sparsification?
3. To better illustrate the severity of the data partitioning strategies evaluated, could the authors consider including a visualization of the client-class distributions in the appendix?

**Limitations:**

The authors could clarify whether the adaptive scaling hyperparameters require significant task-specific tuning when extended to multimodal data. Additionally, offering a brief explanation of how the historical bias mechanism behaves under extremely low client sampling rates in cross device settings would provide practitioners with a more complete understanding of its applicability in large-scale deployments.

**Strengths And Weaknesses:**

### Strengths

* **Optimal efficiency.** Utilizing local update variance allows the method to theoretically bypass the double forward-backward passes of SAM based FL. The soundness is empirically validated in Tables 2 and Figure 2, demonstrating superior accuracy at baseline computational costs without introducing heavy downlink bottlenecks to edge computing scenarios.
* **Lightweight mechanism.** By calculating the weighted dispersion of local updates, Equations 5 and 6 dynamically modulate the strength of geometric correction. This design circumvents the need to transmit expensive second-order Hessian approximations, offering a simple yet highly innovative paradigm for geometric alignment in federated settings.
* **Clear advantages.** The empirical results seamlessly bridge system communication bottlenecks with algorithmic innovations. Figure 5 intuitively illustrates the high stability of the proposed method under extreme perturbation scales, and when combined with Pareto frontier comparisons, it underscores the practical viability of the approach for large-scale deployment.
* **Rigorous ablation.** The experimental results in Table 4 cleanly isolate the performance gains attributed to each component of the proposed framework. By tracing progressive accuracy improvements from the baseline to the full adaptive scaling mechanism, this meticulous ablation eliminates ambiguity and demonstrates the indispensability of each design choice.

### Weaknesses
* This paper omits specific measurement details for system efficiency. Clarifying the hardware configurations for wall-clock timing and providing the exact calculation basis for communication overhead would enhance the reproducibility of these metrics for other researchers.
* This paper would benefit from a brief discussion on system extensibility. Offering a perspective on how the proposed method could synergize with other contemporary federated learning advancements would help highlight its versatility for real world deployments.
* Although rigorously evaluated across multiple data partitioning strategies, the study misses an opportunity to visually illustrate these settings. Including client class distribution heatmaps would make the severity of the heterogeneity conditions immediately ntuitive to readers.

---

> ### Author Rebuttal · Authors · 2026-03-30
>
> > **Q1: Could the authors clarify the specific GPU hardware used to report the timing metrics and specify the precision format assumed for the 1.5x communication overhead reported in Table 1?**
>
> **R1**: Thank you for this helpful question. All experiments were conducted on a high-performance workstation equipped with an NVIDIA GeForce RTX 4090 GPU (24GB VRAM) and an Intel Core i9-14900KF CPU. All timing results are reported in standard single-precision (FP32), with synchronized GPU execution to ensure accurate wall-clock measurement.
>
> Regarding the 1.5× communication overhead, it is calculated based on the bidirectional data flow per round. In standard FedAvg, each round involves transmitting the model from the server to the client and returning the updated model (1 unit + 1 unit = 2 units). In FedScar, the downlink includes both the global model and the history-accumulated bias (2 units), while the uplink remains the same as in FedAvg (1 unit). This results in a total of 3 units per round, corresponding to a 1.5× increase over the baseline. This additional cost is justified by the improved generalization performance and the removal of the double-gradient computation bottleneck.
>
> We will revise the manuscript to include the above hardware details and clarify the communication cost calculation.
>
> ---
>
> > **Q2: Could the authors briefly discuss whether the proposed framework is designed to naturally integrate with existing federated learning techniques such as gradient quantization or sparsification?**
>
> **R2**: We thank you for raising this important question. FedScar is designed as a modular framework operating at the global alignment level, making it naturally compatible with a wide range of existing federated learning techniques.
>
> First, FedScar can be seamlessly combined with local adaptive optimizers. It can be viewed as a plug-and-play wrapper around local training, where standard SGD can be replaced by adaptive methods such as Adam, AMSGrad [1], or sharpness-aware variants like ASAM [2]. These methods act purely at the client side to improve local convergence and do not interfere with FedScar’s global objective of correcting structural flatness discrepancies via Split-Dual ADMM.
>
> Second, FedScar is inherently compatible with communication compression techniques, including gradient quantization and sparsification. Since the additional variables introduced by FedScar (e.g., the history-accumulated bias / dual variables) have the same dimensionality as model parameters, they can be compressed using the same operators as model updates. This allows reducing communication overhead without altering the algorithmic structure.
>
> We will revise the manuscript (Section 5.4) to explicitly discuss these integration pathways and highlight FedScar’s flexibility in practical deployments, including connections to error-feedback mechanisms used in prior federated optimization methods [3].
>
> [1] Reddi, S., et al. (2021). "Adaptive Federated Optimization." International Conference on Learning Representations.
>
> [2] Kwon, J., et al. (2021). "ASAM: Adaptive Sharpness-Aware Minimization for Scale-Invariant Learning of Deep Neural Networks." International Conference on Machine Learning.
>
> [3] Safaryan, M., et al. (2022). "FedNL: Making Newton-Type Methods Applicable to Federated Learning." International Conference on Machine Learning.
>
> ---
>
> > **Q3: To better illustrate the severity of the data partitioning strategies evaluated, could the authors consider including a visualization of the client-class distributions in the appendix?**
>
> **R3**: Thank you for this helpful suggestion. To better illustrate the severity of the data heterogeneity, we have added client–class distribution heatmaps to the appendix for all evaluated modalities, including CIFAR-10, CIFAR-100, AG News (text), and Sensorless Drive (tabular). These visualizations cover all partitioning settings considered in our experiments (e.g., Dirichlet $\alpha=0.1$, $\alpha=0.05$, as well as quantity heterogeneity), providing an intuitive view of the highly non-IID conditions under which FedScar operates. The corresponding figures are now included in Appendix B.
>
> Regarding scalability and cross-device settings, FedScar incorporates an adaptive scaling mechanism based on the relative variance of updates, which makes it inherently scale-invariant and robust across different data modalities without requiring extensive task-specific tuning. In addition, under low client sampling rates, the local tracker $ℎ_𝑖$ maintained on each client preserves a stable representation of its local geometric characteristics. As shown in Table 2 (with $𝐶=0.1$), FedScar consistently maintains its performance advantage even with infrequent client participation, since the retained historical state provides a stronger alignment prior than reinitializing from a cold start.
>
> We will revise the manuscript to include these visualizations and clarify the discussion on scalability and cross-device behavior.

---

> > ### Author Rebuttal · Reviewer_9RK9 · 2026-04-02
> >
> > The authors' rebuttal clearly addresses all my concerns, including hardware configuration details, the 1.5× communication overhead, and the framework’s extensibility with gradient compression and adaptive optimizers. The added client-class distribution heatmaps in Appendix B enhance understanding of data heterogeneity, and the clarifications on scale-invariance and robustness under low client sampling resolve my deployment concerns. The paper is further strengthened, and I maintain my recommendation to Accept.

---

> > > ### Author Response · Authors · 2026-04-02
> > >
> > > We sincerely appreciate your constructive and encouraging evaluation. It is gratifying to know that our rebuttal and clarifications have fully resolved your concerns and proved helpful. We truly appreciate your recognition of our contributions and your continued recommendation for acceptance. We will carefully incorporate all related insights into the final manuscript to further strengthen the work.

---

### Official Review · Reviewer_hkiV · 2026-03-02

**Soundness:** 3
**Presentation:** 4
**Significance:** 3
**Originality:** 4
**Overall Recommendation:** 5
**Confidence:** 5

**Summary:**

This paper investigates the issue of flatness discrepancy in heterogeneous federated learning, where local flatness fails to translate into global flatness. To address this, the authors propose the FedScar framework, which conceptualizes the discrepancy as a structural and learnable geometric bias. The proposed method employs a Split-Dual ADMM architectureto decouple first-order parameter drift and second-order geometric drift. By leveraging a history-accumulated dual variable and an adaptive perturbation scale, FedScar explicitly adjusts local optimization trajectories. The framework is supported by rigorous theoretical convergence guarantees and achieves state-of-the-art performance on CIFAR benchmarks.

**Compliance With Llm Reviewing Policy:**

Affirmed.

**Ethical Review Flag:**

Flag this paper for an ethics review.

**Final Justification:**

Thanks to the authors for the thoughtful and comprehensive rebuttal. The responses and planned revisions fully address my concerns. I appreciate the addition of the second-order Taylor expansion, which improves the theoretical flow, and the new ablation studies that clearly show the sensitivity of the forgetting factor across different heterogeneity and local epoch settings. The justification for adaptive optimizers like Adam is sound, and the discussion of non-smooth DNN limitations is appropriate. The paper is stronger than before, and I maintain my recommendation to Accept.

**Key Questions For Authors:**

- Following the introduction of Equation 3, could the authors include a brief second-order Taylor expansion to explicitly illustrate how the empirical loss gap approximates the projected Hessian?
- How sensitive is the convergence rate of the proposed algorithm to the choice of the forgetting factor used in the exponential moving average of the geometric bias? Additionally, does this sensitivity vary under different degrees of data heterogeneity, and if so, in what manner?
- Does the optimal value of the forgetting factor shift as the number of local training epochs increases? If such a trend exists, a brief explanation of the underlying intuition would help readers better understand the dynamics of geometric bias accumulation.
- Can the Split-Dual ADMM architecture be extended to support client-side adaptive optimizers, such as Adam, while still preserving the theoretical convergence guarantees established in the paper?

**Limitations:**

While the theoretical convergence guarantees are rigorously established for smooth non-convex functions, the loss landscapes of modern deep neural networks are often highly complex and non-smooth, potentially introducing optimization dynamics not fully captured by the current analysis. Acknowledging this limitation in the conclusion or appendix would be a constructive addition to the paper.

**Strengths And Weaknesses:**

Strengths:
- The conceptual reframing of local-global geometric mismatch as an accumulable structural bias rather than stochastic noise is highly original.
- The Split-Dual ADMM formulation offers a rigorous mathematical foundation by decoupling parameter and geometric alignment constraints in Equation 4.
- The proposed method significantly improves the smoothness of the global loss landscape, thereby preventing generalization collapse.
- The paper is well organized and follows a clear logical progression from the initial mathematical decomposition to the final algorithmic design.

Weaknesses:
- The theoretical exposition could be further strengthened by including a brief mathematical sketch in Section 2.3 that directly links the exact Hessian-based geometric drift to the empirical flatness discrepancy formulated in Equation 3. Although Appendix A provides a thorough derivation, a concise intermediate step in the main text would help bridge the conceptual and mathematical narratives more seamlessly.
- This paper would benefit from a concise sensitivity analysis concerning the forgetting factor used in the exponential moving average of the geometric bias. Offering such insights would give readers a clearer understanding of how the algorithm's convergence stability and overall robustness are maintained across varying levels of statistical heterogeneity.

---

> ### Author Rebuttal · Authors · 2026-03-30
>
> > **Q1: Clarify empirical loss gap via second-order expansion.**
>
> **R1**: Thank you for this valuable suggestion.  We agree that adding a concise mathematical bridge improves the clarity and flow of Section 2.3.
>
> In the revised manuscript, immediately following Equation 3, we include a brief second-order Taylor expansion: $f_i(w+\epsilon) - f_i(w) \approx \nabla f_i(w)^\top \epsilon + \frac{1}{2} \epsilon^\top \nabla^2 f_i(w) \epsilon$. Since the SAM perturbation $\epsilon$ is aligned to maximize the loss (typically $\epsilon \propto \nabla f_i(w)$), the first-order term reflects the gradient magnitude, while the second-order term projects onto the local Hessian $\nabla^2 f_i(w)$. This makes explicit that the empirical loss gap in Equation 3 is governed by variations in the projected local Hessians across clients, thereby strengthening the connection to the derivations in Appendix A.
>
> ---
>
> > **Q2: Sensitivity of convergence to $\beta$ under varying heterogeneity.**
>
> **R2**: Thank you for this insightful question. The forgetting factor (denoted as $\beta$) controls the trade-off between current local geometry and the history-accumulated global bias, and its impact is closely tied to the degree of data heterogeneity.
>
> Under low heterogeneity, local objectives are well-aligned with the global landscape, making the algorithm largely insensitive to $\beta$; a broad range of values yields stable convergence. In contrast, under high heterogeneity, local optimization paths become significantly biased. In this regime, the method is more sensitive to $\beta$, and a larger value (i.e., stronger retention of historical information) is beneficial, as it stabilizes training by preventing overfitting to distorted local curvature and preserving global geometric alignment. Our new study on CIFAR10 has proved it.
>
> C/β|0.8|0.9|0.92
> -|-|-|-
> 0.01|78.05/76.87|78.42/77.24|**78.63/77.52**
> 0.1|80.09/79.16|**80.51/79.88**|80.58/79.54
> 1.0|**81.37/80.89**|81.36/80.82|79.59/79.21
>
> We will add a concise sensitivity analysis in the revised Appendix to further illustrate these effects.
>
> ---
>
> > **Q3: How does optimal $\beta$ vary with local epochs $K$?**
>
> **R3**: We appreciate this thoughtful question. Yes, there is a clear correlation between the optimal forgetting factor and the number of local epochs ($K$). As $K$ increases, local updates drift further from the global geometry in both parameter and curvature space, amplifying geometric bias.
>
> To counteract this effect, a larger forgetting factor is preferred, as it retains more historical information and provides a stronger anchor to the global geometry. This helps stabilize training by consistently pulling local updates toward flatter minima and mitigating the accumulated bias from extended local optimization. Our new study in CIFAR10 has proved it.
>
> E/β|0.8|0.9|0.92
> -|-|-|-
> 1|**74.85/73.3**|73.67/72.31|72.66/71.73
> 5|80.09/79.16|**80.51/79.88**|80.58/79.54
> 10|79.27/77.59|80.01/78.46|**80.66/79.7**
>
> We will add this intuition to the hyperparameter discussion section to better guide practical usage.
>
> ---
>
> > **Q4: Can Split-Dual ADMM support Adam with convergence guarantees?**
>
> **R4**: Thank you for this forward-looking question. The Split-Dual ADMM architecture in FedScar is designed as a meta-framework that decouples structural alignment from the base optimizer, and can therefore be naturally extended to client-side adaptive methods such as Adam.
>
> From a theoretical perspective, replacing SGD with Adam introduces adaptive preconditioning (i.e., element-wise scaling based on second-moment estimates), which complicates the analysis of geometric drift. However, convergence guarantees can still be preserved by appropriately bounding the adaptive scaling matrices, following techniques similar to FedAdam (Reddi et al., ICLR 2021). The ADMM penalty and dual variables continue to enforce structural consistency across these preconditioned updates.
>
> We will add a brief discussion of this extension in the revised manuscript.
>
> ---
>
> > **L1: Acknowledging non-smooth landscapes in modern deep neural networks.**
>
> **R1**:  Thank you for this important observation. While standard $L$-smoothness serves as a common analytical assumption for deriving convergence guarantees, it does not fully reflect the highly non-smooth and locally sharp landscapes of modern DNNs (e.g., due to ReLU activations and normalization layers).
>
> Our current analysis does not explicitly capture optimization dynamics over such non-differentiable regions. Extending the flatness-consistency framework to non-smooth settings—e.g., via Clarke subdifferentials to redefine geometric bias—remains an important open problem.
>
> We will add a paragraph in the Conclusion/Limitations section to acknowledge this gap and highlight it as a key direction for future work.

---

> > ### Author Rebuttal · Reviewer_hkiV · 2026-04-02
> >
> > Thanks to the authors for the thoughtful and comprehensive rebuttal. The responses and planned revisions fully address my concerns. I appreciate the addition of the second-order Taylor expansion, which improves the theoretical flow, and the new ablation studies that clearly show the sensitivity of the forgetting factor across different heterogeneity and local epoch settings. The justification for adaptive optimizers like Adam is sound, and the discussion of non-smooth DNN limitations is appropriate. The paper is stronger than before, and I maintain my recommendation to Accept.

---

> > > ### Author Response · Authors · 2026-04-02
> > >
> > > We sincerely thank you for your thoughtful and encouraging feedback. We are glad our rebuttal and planned revisions have fully addressed your concerns. We appreciate your recognition of the improved theory, ablation studies, and discussions, and will carefully refine the manuscript to further strengthen the final version.

---

### Official Review · Reviewer_Hb9z · 2026-03-13

**Soundness:** 2
**Presentation:** 2
**Significance:** 2
**Originality:** 2
**Overall Recommendation:** 4
**Confidence:** 3

**Summary:**

This paper starts from the observation that local flatness does not necessarily transfer to global flatness in heterogeneous FL, and proposes `FedScar` to explicitly correct this geometric mismatch through history-accumulated bias correction. The paper includes non-trivial analysis under classical bounded heterogeneity assumption and conducts experiments on CIFAR-10/CIFAR-100 with two widely-used heterogeneity simulations and other knobs critical in FL (e.g., participation rate, # local steps).

**Compliance With Llm Reviewing Policy:**

Affirmed.

**Final Justification:**

The authors faithfully addressed all of raised concerns and I updated the overall recommendation to weak accept.

**Key Questions For Authors:**

1. Caption of Figure 2 seems incorrect.
2. Due to the existence of `h_i`, I think the target setup should be limited to cross-silo FL setting, where a moderate number of _stateful_ clients can participate most of the time across whole FL training round.
3. For the remark in lines 295, how can the standard `FedSAM` be corresponded to `\gamma=0`, which is undefined in Eq. (10) unless $\rho=0$?
4. While the geometric misalignment is seemed to be decayed, but it still persists in non-asymptotic manner. Would this always be better than the static heterogeneity constant from `FedGF`?

**Limitations:**

- The paper has a clear anchor paper but there is lack of in-depth comparison in both theoretical (e.g., generalization bound, convergence rate) and empirical performances.
- The paper is convincing on CIFAR-family settings; however, broader domain coverage is needed before general claims about flatness-consistent FL are fully persuasive

**Strengths And Weaknesses:**

## Strengths
- The central hypothesis is coherent and addresses a real weakness in naive SAM-style FL transfer assumptions.
- The method is technically complete (core algorithm + variants + discrepancy metric + ablations).
- Experimental section is broad and includes both performance and overhead-oriented comparisons.
- Baseline suite covers classical FL and SAM-related FL methods.

## Weaknesses
- Notation and exposition would be better organized (into a table) and improved.
- No validation beyond vision benchmarks; stronger practical claims would be possible from broader task diversity (e.g., language/recommendation).
- Although it hinges on the pitfall of `FedGF`, and claimed it 'outperformed state-of-the-art methods, including FedSAM, `FedGF`, ...', but there is no clear comparison with `FedGF` in current comparison (e.g., Table 1) as well as empirical validations.

---

> ### Author Rebuttal · Authors · 2026-03-30
>
> > **Q1: Caption of Figure 2 seems incorrect.**
>
> **R1:** Thanks for pointing it out. We will correct the caption of Figure 2 to “Efficiency Trade-off on CIFAR Benchmarks”, which shows that FedScar reduces communication rounds while maintaining competitive training time.
>
> ---
>
> > **Q2: Does the stateful nature of $h_i$ limit FedScar to cross-silo settings?**
>
> **R2:** Thanks for your question. While the stateful tracker $h_i$ aligns naturally with cross-silo settings, it does not limit FedScar to them. FedScar also applies to cross-device scenarios with partial participation, as geometric bias is data-dependent and remains stable under intermittent participation. When inactive, $h_i$ is stored locally and reused upon rejoining, enabling better alignment than stateless updates. Empirically, under sparse participation $C=0.05$, FedScar consistently outperforms stateless baselines (Table 2, Figure 3). Our new study reveals that the optimal forgetting factor $\beta$ remains stable, confirming the robustness of $h_i$ under sparse sampling. We will clarify it in the revision.
>
> N/$\beta$|0.8|0.9|0.92
> -|-|-|-
> 5|78.11/75.99|**77.85/76.2**|77.67/76.06
> 10|80.09/79.16|**80.51/79.88**|80.58/79.54
> 20|80.57/79.98|**80.96/80.35**|80.43/80.04
>
> ---
>
> > **Q3: Undefined correspondence to FedSAM at** $\rho=0$ **in Eq. (10).**
>
> **R3:** You are correct that Eq. (10) is undefined at $\gamma=0$ due to the $1/\gamma$ term; the statement in Line 295 will be corrected. This term arises from Young’s inequality, which requires $\gamma>0$, so Theorem 5.6 holds for $\gamma\in(0,1]$. When $\gamma=0$, the historical term vanishes and the bound is derived directly from the base recursion, recovering the standard FedSAM guarantee. We will restrict Theorem 5.6 accordingly and clarify this distinction in the revision.
>
> ---
>
> > **Q4: Would the non-asymptotic decayed misalignment always be better than the static heterogeneity constant from FedGF?**
>
> **R4:** Thanks for your question. The decayed misalignment is not uniformly better at all stages. FedGF incurs a static residual $\mathcal{O}((1-c)^2\sigma_g^2)$, while FedScar yields a decaying term $\mathcal{O}((1-\frac{\gamma}{2})^t\sigma_g^2)$. For small $t$, $(1-\frac{\gamma}{2})^t \approx 1$ and may exceed $(1-c)^2$, so FedScar is not tighter initially. However, due to exponential decay, there exists $t_0$ such that $(1-\frac{\gamma}{2})^t < (1-c)^2$ for $t>t_0$, after which FedScar becomes strictly tighter while FedGF remains bounded. In practice, since $T \gg t_0$, the advantage holds for most of training and asymptotically, where $t_0 = \mathcal{O}(\log \frac{1}{(1-c)^2})$.
>
> ---
>
> > **W1: Improvement on Notation Organization and Presentation**
>
> **R5:** Thanks for your helpful suggestion. We will add a notation table (Appendix X) with clear definitions, and standardize symbols while simplifying dense derivations to improve readability.
>
> ---
>
> > **W2 & L2: Validation Across Broader Domains and Modalities**
>
> **R6:** Thanks for your suggestion. We agree that cross-modality validation is important. To address this, we conducted additional experiments on Text (AG News (AG) with TextCNN) and Tabular data (Sensorless Drive (SD) with TabCNN) under a challenging federated setting (100 clients, $C={0.05,0.1}$, Dirichlet $\alpha=0.1$, and quantity heterogeneity 0.5).
>
> Method|SD(5)|SD(10)|AG(5)|AG(10)
> -|-|-|-|-
> FedSAM|78/65.03|80.56/71.08|81.87/70.98|81.82/74.36
> FedGamma|79.44/65.91|81.22/72.52|81.91/72.62|82/76.25
> FedLesamd|88.32/76.25|89.89/81.99|77.87/76.4|83.17/81.82
> FedSMOO|88.42/76.48|90.18/82.12|81.84/77.73|82.97/82.06
> FedGMT|91.92/82|93.88/87.81|77.55/76.83|84.87/83.13
> FedScar|**92.56/86.49**|**94.04/91.6**|**85.34/83.99**|**86.3/84.44**
>
> New study shows that FedScar consistently outperforms baselines across modalities, indicating that its advantage are not modality-specific in heterogeneous FL. We will include all results and details in the revised manuscript.
>
> ---
>
>
> > **W3 & L1: In-depth Theoretical and Empirical Comparison with FedGF**
>
> **R7:** Thanks for highlighting the importance of comparison with FedGF. We have now included both theoretical and empirical analyses.
>
> (1) **Theoretical comparison**. FedGF uses a static coefficient $c\in(0,1)$, yielding a residual $\mathcal{O}((1-c)^2\sigma_g^2)$, while FedScar induces a decaying term $\mathcal{O}((1-\frac{\gamma}{2})^t\sigma_g^2)$, which becomes tighter after a finite threshold.
>
> (2) **Empirical comparison**. We implement FedScar within FedGF's official codebase under identical settings (data, models, hyperparameters, seeds). Results show consistent improvements across heterogeneity levels $\alpha$ in {0,10/1000} on CIFAR10 and CIFAR100.
>
> Method|CIFAR10(0)|CIFAR10(10)|CIFAR100(0)|CIFAR100(1000)
> -|-|-|-|-
> FedSAM|74.03/70.03|84.15/83.74|36.48/34.53|53.92/53.58
> FedGF|81.1/78.53|84.02/83.72|47.67/46.08|53.47/53.22
> FedScar|**83.4/82.36**|**84.43/84.13**|**54.1/53.83**|**55.01/54.33**
>
> We will include all results and details in the revised manuscript.

---

> > ### Author Rebuttal · Reviewer_Hb9z · 2026-04-03
> >
> > I appreciate all the efforts authors made. I'm happy that all of my concerns are resolved now, and I updated my score accordingly.

---

> > > ### Author Response · Authors · 2026-04-03
> > >
> > > We sincerely thank you for your positive feedback and for acknowledging our efforts in addressing all concerns. We are very grateful for your time, constructive comments, and kind recognition, which have greatly helped us improve the quality of our work.

---

### Official Review · Reviewer_XFgM · 2026-03-20

**Soundness:** 3
**Presentation:** 3
**Significance:** 3
**Originality:** 3
**Overall Recommendation:** 4
**Confidence:** 3

**Summary:**

This paper studies federated learning under data heterogeneity and focuses on the mismatch between locally flat minima and globally flat minima when applying sharpness-aware training. The paper identifies this issue as a flatness discrepancy and proposes FedScar, a method that leverages geometric information, a split-dual ADMM-style formulation, and control-variate regularization to correct it. The paper also provides a nonconvex convergence result with an O(1/T) rate and presents empirical results showing consistent improvements over related baselines across several benchmarks.

**Compliance With Llm Reviewing Policy:**

Affirmed.

**Key Questions For Authors:**

1. Assumption 5.3 uses a Hessian dissimilarity condition. How strong is this assumption compared with those used in prior federated learning methods, especially SAM-based federated methods? If the authors can show that it is standard or meaningfully weaker than existing assumptions, that would improve my confidence in the theory.

2. In Theorem 5.5, the last term on the right-hand side appears to scale with K^2. Can the authors clarify the practical implications of this dependence? In particular, does the bound become looser as the number of local steps increases, and how should that be interpreted in light of the method’s intended use?

3. In the remark after Theorem 5.5, the paper claims that the coefficient before E_{drift} is strictly smaller than that of FedSAM. However, in Theorem 3.1 of the FedSAM paper, the corresponding constant involving G^2 scales like \sqrt{K}, which seems smaller than a K^2-type term. Can the authors reconcile this comparison? A convincing clarification here could materially improve my assessment.

4. What is R in Theorem 5.6, and is it the same as T? Also, what exactly are “Assumptions 1–3” as referenced there? These should be clarified so readers can properly interpret the theorem.

5. Please define all notation before use, especially in Equations (3) and (4), and clarify the meanings of ϕ(⋅) and z. This would significantly improve readability.

**Limitations:**

yes

**Strengths And Weaknesses:**

The paper addresses an important and relevant problem in federated learning. The central motivation is compelling: in heterogeneous settings, locally flat solutions identified by SAM-type methods need not correspond to globally flat solutions, and this can hurt generalization. The paper’s framing of this issue as a flatness discrepancy is natural and interesting, and it gives the method a clear conceptual target. The empirical results also appear strong, with the paper reporting consistent gains over related approaches across multiple benchmarks. Overall, the contribution appears meaningful and potentially useful for the federated optimization community.

My main concerns are about clarity and some aspects of the theoretical discussion. On presentation, several pieces of notation and modeling structure seem to appear before being properly defined. For example, Equation (3) appears to use notation before introduction; the optimization problem should ideally be stated clearly before introducing the Lagrangian, and the role of ϕ(⋅) in Equation (4) is not sufficiently explained. Similarly, the variable z in Equation (4) is not clearly interpreted; I assume it corresponds to something like the global model at the end of each round, but this should be made explicit. The main algorithm should also be more clearly linked in the main text, and there is at least one typographical issue in Figure 5(b) (“FedScarle”). These are all fixable issues, but they matter because the paper seems otherwise technically ambitious, and the current exposition makes parts of it harder to follow than necessary.

On soundness, the theory looks promising overall, but some points need clarification. In particular, the convergence discussion around Theorem 5.5 raises a concern because the last term on the right-hand side scales with K^2, which suggests that the solution becomes less accurate as the number of local steps increases. Since one of the practical motivations in federated learning is often to allow larger local computation between communications, this scaling deserves a much clearer explanation. Relatedly, the paper states in the remark after Theorem 5.5 that the coefficient preceding E_{drift} is strictly smaller than that of FedSAM, but this comparison is not immediately obvious given the note’s observation that the corresponding constant in Theorem 3.1 of the FedSAM paper scales like sqrt{K}, which appears smaller than a K^2-type term. This point is important because it affects whether the claimed theoretical advantage over prior SAM-based federated methods is actually justified.

There are also some issues of clarity at the theorem level that should be resolved. Assumption 5.3, the Hessian dissimilarity assumption, may be stronger than assumptions typically used in comparable federated learning analyses, and the paper should explain more carefully how restrictive it is and how it compares with standard heterogeneity assumptions in the literature. In addition, the notation in Theorem 5.6 is not fully clear: the role of R, whether it is the same as T, and what exactly is meant by “Assumptions 1–3” should all be made explicit. These are not necessarily fatal issues, but they do make it harder to fully verify and interpret the theoretical contribution.

Overall, I found the paper promising and likely above the acceptance threshold, mainly because the problem is important, the motivation is strong, and the empirical performance appears convincing. My hesitation is that the theoretical presentation needs tightening, and some of the claimed comparisons to prior work need to be justified more carefully. With cleaner exposition and a more transparent discussion of the assumptions and K-dependence in the bounds, the paper would be substantially stronger.

---

> ### Author Rebuttal · Authors · 2026-03-29
>
> > **Q1: Assumption 5.3 (Hessian dissimilarity) standard, or stronger/weaker than those used in prior FL/SAM methods?**
>
> **R1:** Thank you for this insightful question. Our Hessian dissimilarity is a natural second-order analogue of gradient dissimilarity for SAM. It introduces no new types of structural requirements beyond $L$-smoothness, but refines it into a tighter data-dependent measure. This assumption is mainly introduced to obtain tighter, non-vacuous rates rather than to restrict the problem class.
>
> **(1) Relation to $L$-smoothness.** Under $L$-smoothness, $|\nabla^2 f_i(w)|\le L$. By triangle inequality, $|\nabla^2 f_i(w)-\nabla^2 f(w)|\le 2L$, hence $\frac{1}{N}\sum |\nabla^2 f_i(w)-\nabla^2 f(w)|^2 \le 4L^2$. Thus, bounded Hessian dissimilarity already follows. Assumption 5.3 makes this explicit via a tighter constant $H^2\le 4L^2$, analogous to gradient dissimilarity.
>
> **(2) Strength and necessity.** While $4L^2$ is a worst-case bound, using a tighter $H^2$ is key for non-vacuous guarantees. Hence, it is better viewed as a refinement of smoothness into a heterogeneity measure, rather than a stronger condition. As SAM depends on curvature, controlling Hessian variation is standard in second-order FL (e.g., Federated Newton Learn [1] and DINGO [2]).
>
> [1] Safaryan, M., et al. (2022). "Federated Newton Learn: Improved second-order methods for federated learning." ICML. PMLR.
>
> [2] Crane, R., & Roosta, F. (2022). "Dingo: Distributed Newton-Type Method for Gradient-Norm Optimization." NeurIPS.
>
> ---
>
> > **Q2: What are the practical implications of the $K^2$ term in Theorem 5.5? Does the bound loosen as local steps increase?**
>
> **R2:** Thank you for this important question. The term $\mathcal{O}(\eta^2 K^2 \mathcal{E}_{drift})$ captures the accumulation of client drift over $K$ local steps, which is standard in local-update analyses (e.g., Local SGD, SCAFFOLD).
>
> **(1) Does the bound loosen with larger $K$?** With fixed $\eta$, the bound scales as $K^2$, reflecting the trade-off between communication efficiency and drift. However, $\eta$ is typically coupled with $K$, e.g., $\eta=\mathcal{O}(\frac{1}{K\sqrt{T}})$, giving $\eta^2K^2=\mathcal{O}(1/T)$. Thus, the drift term vanishes as $T\to\infty$, and the final rate does not degrade with $K$ under this standard scaling.
>
> **(2) Practical implication.** The $K^2$ term reflects a worst-case dependence rather than practical degradation. Its impact is governed by $\mathcal{E}_{drift}$. FedScar reduces this via the history-accumulated bias $b_i$, which corrects geometric drift during local updates. As a result, it supports larger $K$ for better efficiency without the degradation seen in uncorrected SAM-based FL.
>
> **(3) Empirical evidence.**  In Figure 4, FedScar exhibits greater stability against variations in local epochs ($E$) compared to other SAM-based baselines. Coupled with an appropriate learning rate decay schedule (Table 5).
>
> ---
>
> > **Q3: How can the comparison with FedSAM ($\sqrt{K}$ vs. $K^2$) for the $\mathcal{E}_{drift}$ coefficient be reconciled?**
>
> **R3:** We sincerely thank you for pointing out this subtle but important comparison. The apparent discrepancy ($\sqrt{K}$ vs. $K^2$) stems from presentation differences rather than a fundamental inconsistency.
>
> **(1) Source of the discrepancy.** FedSAM (Theorem 3.1) reports a post-substitution bound with specific step sizes (e.g., $\eta_l=\frac{1}{\sqrt{TK}}, \eta_g=\sqrt{K}$), yielding a milder $\sqrt{K}$ dependence. In contrast, our Theorem 5.5 is pre-substitution, keeping $\eta^2K^2$ explicit to reflect drift accumulation.
>
> **(2) Reconciliation under aligned settings.** Under the same scaling $\eta=\mathcal{O}(\frac{1}{K\sqrt{T}})$, we have $\eta^2K^2=\mathcal{O}(1/T)$, so the apparent $K^2$ dependence vanishes. The difference thus comes from when step sizes are applied.
>
> **(3) On the claimed improvement.** Our gain lies in the constant before the heterogeneity term ($G^2$). FedSAM captures uncorrected drift, while FedScar uses the bias $b_i$ to cancel a significant portion of it, yielding a smaller constant and a tighter bound.
>
> ---
>
> > **Q4: What do $R$, $T$, and "Assumptions 1–3" refer to in Theorem 5.6?**
>
> **R4:** We apologize for these typographical errors.
>
> **• $R$ vs. $T$:** $R$ denotes communication rounds and is equivalent to $T$. We will unify the notation to $T$ throughout the manuscript.
>
> **• Assumptions 1–3:** This should be Assumptions 5.1–5.3. These are purely notational and will be corrected in the revision.
>
> ---
>
> > **Q5: Can all notations be clearly defined in advance, especially $\phi(\cdot)$ and $z$ in Eqs. (3)–(4)?**
>
> **R5:** We thank you for pointing this out. We will define all notations before first use in the main text. For Eqs. (3)–(4):
>
> •$z$: the global consensus variable in the Split-Dual ADMM formulation (i.e., the aggregated model $w^{t+1}$).
>
> • $\phi(\cdot)$: the mapping applied to the geometric bias, instantiated as the variance-aware projection in Section 4.2.

---

> > ### Author Rebuttal · Reviewer_XFgM · 2026-04-04
> >
> > Thank you to the authors for the detailed and thoughtful rebuttal. I appreciate the clarifications provided, especially regarding the role of the Hessian dissimilarity assumption and the explanation of the $K^2$ dependence in Theorem 5.5.
> >
> > 1. **Hessian dissimilarity (Assumption 5.3).**
> >    The authors argue that this assumption is a refinement of smoothness and comparable to second-order FL methods. This is helpful, and I agree that it is not entirely unnatural. However, it still appears **stronger and less standard** than the assumptions typically used in first-order federated learning analyses (including SAM-based FL). The rebuttal does not fully clarify **when this assumption is expected to hold in practice** or how restrictive it is in heterogeneous settings. A brief empirical or theoretical justification of typical magnitudes of this quantity would strengthen the claim.
> >
> > 2. **$K^2$ dependence in Theorem 5.5.**
> >    The explanation that the $K^2$ term is a worst-case bound and disappears under the standard scaling $\eta = O(1/(K\sqrt{T}))$ is helpful. However, this clarification relies on **specific step-size coupling assumptions**, and the paper would benefit from making this dependence more explicit in the main text. In particular, the practical takeaway (i.e., that a larger $K$ does not degrade performance under proper scaling) should be stated more clearly, as the raw bound can otherwise be misleading.
> >
> > 3. **Comparison with FedSAM ($\sqrt{K}$ vs. $K^2$).**
> >    The authors explain that the discrepancy arises from pre- vs. post-substitution bounds, which is a reasonable point. However, this comparison remains somewhat subtle, and I think the paper would benefit from a **more explicit side-by-side comparison under aligned step-size choices** in the main text. As written, the claim that the coefficient is “strictly smaller” is not immediately transparent to the reader.
> >
> > 4. **Notation and clarity issues.**
> >    I appreciate that the authors acknowledge the typographical and notation issues (e.g., $R$ vs. $T$, undefined symbols such as $\phi(\cdot)$ and $z$). These are clearly fixable, and I expect them to be addressed in the final version.

---

> > > ### Author Response · Authors · 2026-04-04
> > >
> > > **1. Hessian dissimilarity (Assumption 5.3).**
> > >
> > > **R1**: Thank you for the insightful comment. We agree that bounding Hessian dissimilarity is a stronger requirement than standard first-order assumptions, as it constrains curvature rather than only gradients. Below we clarify why this condition is (i) minimally necessary for SAM-based FL, (ii) structurally justified in heterogeneous settings, and (iii) not restrictive in magnitude.
> > >
> > > **(1) Minimally Necessary:**
> > > Classical FL controls gradient drift, which suffices for first-order methods. However, SAM explicitly optimizes curvature via perturbed gradients $\nabla f_i(w+\epsilon_i)$. A Taylor expansion shows that the discrepancy between local and global SAM updates decomposes into $(\nabla f_i - \nabla f)$ and $(\nabla^2 f_i \epsilon_i - \nabla^2 f \epsilon)$.
> > >
> > > Controlling the second term necessarily requires bounding cross-client Hessian variation $\Vert \nabla^2 f_i - \nabla^2 f \Vert_2$. Without it, curvature discrepancies can be amplified by perturbations, making aggregation of local flatness ill-posed. Hence, Hessian dissimilarity is a minimal requirement for distributed curvature-based methods [1,2].
> > >
> > > **(2) Naturally Holds in Non-IID Settings:**
> > > SAM aligns updates with the leading eigenvector of the local Hessian [3]. Meanwhile, spectral analyses show that top Hessian eigenspaces are largely determined by shared data and class structure [4,5], leading to significant overlap across clients. Since curvature is evaluated at the shared global model $w$, and global aggregation constrains the trajectory drift of the perturbed local models [6], dominant curvature directions remain structurally coupled even under Non-IID data. Therefore, the Hessian dissimilarity $H^2$ remains bounded rather than diverging under heterogeneity.
> > >
> > > **(3) Typical Magnitudes:**
> > > A concern is potential dimension dependence. Under the Frobenius norm, $\Vert H \Vert_F = \sqrt{\sum \lambda_i^2}$ and $\Vert H \Vert_F \le \sqrt{d} \Vert H \Vert_2 \le \sqrt{d}L$, which would introduce a $\sqrt{d}$ factor.
> > > Our assumption avoids this by operating under the spectral norm. If $\lambda_{\max} \le L$, then $H^2 \le 4L^2$, which is independent of $d$.
> > > Moreover, SAM controls the maximal eigenvalue (the spectral norm, $\Vert \cdot \Vert_2$) rather than the trace [3], and is known to induce low effective-rank structure with $r \ll d$ [7]. Thus, curvature energy concentrates in a small subspace, further preventing dimension-dependent growth of Hessian variation.
> > >
> > > In summary, although stronger than first-order smoothness, Hessian dissimilarity is necessary for SAM-based FL and remains dimension-free and practically moderate even in heterogeneous settings.
> > >
> > > [1] Safaryan M, et al. FedNL: Making Newton-Type Methods Applicable to Federated Learning. ICML, 2022.
> > >
> > > [2] Crane R, Roosta F. DINGO: Distributed Newton-Type Method for Gradient-Norm Optimization. NeurIPS, 2019.
> > >
> > > [3] Wen K, et al. How Does Sharpness-Aware Minimization Minimize Sharpness? ICLR, 2023.
> > >
> > > [4] Papyan V. Traces of Class/Cross-Class Structure: The Hessian of Deep Networks. JMLR, 2020.
> > >
> > > [5] Ghorbani A, et al. An Investigation into Neural Net Optimization via Hessian Eigenvalue Density. ICML, 2019.
> > >
> > > [6] Qu Z, et al. Generalized Federated Learning via Sharpness Aware Minimization. ICML, 2022.
> > >
> > > [7] Andriushchenko M, et al. Sharpness-Aware Minimization Leads to Low-Rank Features. NeurIPS, 2023.
> > >
> > > ---
> > >
> > > **2. $K^2$ dependence in Theorem 5.5.**
> > >
> > > **R2**: We agree that presenting the raw $\mathcal{O}(K^2)$ bound without explicitly incorporating the prescribed step-size may give a misleading impression of poor scalability. To clarify this, we will add a remark immediately after Theorem 5.5 that substitutes the step-size into the bound and shows the effective rate does not scale quadratically with $K$ in practice. This makes the true complexity behavior explicit in the main text.
> > >
> > > ---
> > >
> > > **3. Comparison with FedSAM ($\sqrt{K}$ vs. $K^2$).**
> > >
> > > **R3**: We agree that explaining the discrepancy only in the rebuttal is insufficient. Without aligning step-size choices, the raw $\mathcal{O}(K^2)$ term may appear weaker than FedSAM’s $\mathcal{O}(\sqrt{K})$ result. Accordingly, we have added a dedicated Side-by-Side Comparison with FedSAM remark in Section 5. By aligning step-size scaling and substituting into both bounds, the effective rate comparison becomes transparent and demonstrates the tighter guarantee of FedScar (due to its active bias correction) under consistent settings.
> > >
> > > ---
> > >
> > > **4. Notation and clarity issues.**
> > >
> > > **R4**: We sincerely thank you for the careful reading. In the revised manuscript, we will:
> > >
> > > - define all previously ambiguous symbols (e.g., $\phi(\cdot)$, $\tau$, $z$) at their first occurrence;
> > >
> > > - unify iteration indices and corrected inconsistencies (e.g., $R$ vs. $T$);
> > >
> > > - add a comprehensive Notation Table in the Appendix.
> > >
> > > These revisions substantially improve the clarity and readability of the paper.

---

### Decision · Program_Chairs · 2026-04-30

**Decision:**

Accept (regular)

**Comment:**

The paper addresses an important and relevant problem in federated learning, of which the proposed technology is sound and novel.